# Base-editing corrects metabolic abnormalities in a humanized mouse model for glycogen storage disease type-Ia

Irina Arnaoutova[1], Yvonne Aratyn-Schaus[2], Lisa Zhang[1], Michael S. Packer[2], Hung-Dar Chen[1], Cheol Lee[1], Sudeep Gautam[1], Francine M. Gregoire[2], Dominique Leboeuf[2], Steven Boule[2], Thomas P. Fernandez[2], Victoria Huang[2], Lo-I Cheng [2], Genesis Lung [2], Brianna Bannister[2], Jeremy Decker[2], Thomas Leete [2], Lan S. Shuang[2], Caroline Bock[2], Prachi Kothiyal[2], Phil Grayson[2], Ka W. Mok[2], Jeffrey J. Quinn [2], Lauren Young[2], Luis Barrera [2], Giuseppe Ciaramella [2], Brian C. Mansfield [1] & Janice Y. Chou [1] ✉

Glycogen storage disease type-Ia patients, deficient in the *G6PC1* gene encoding glucose-6-phosphatase-α, lack blood glucose control, resulting in life-threatening hypoglycemia. Here we show our humanized mouse model, huR83C, carrying the pathogenic *G6PC1*-R83C variant displays the phenotype of glycogen storage disease type-Ia and dies prematurely. We evaluate the efficacy of BEAM-301, a formulation of lipid nanoparticles containing a newly-engineered adenine base editor, to correct the *G6PC1*-R83C variant in huR83C mice and monitor phenotypic correction through one year. BEAM-301 can correct up to ~60% of the *G6PC1*-R83C variant in liver cells, restores blood glucose control, improves metabolic abnormalities of the disease, and confers long-term survival to the mice. Interestingly, just ~10% base correction is therapeutic. The durable pharmacological efficacy of base editing in huR83C mice supports the development of BEAM-301 as a potential therapeutic for homozygous and compound heterozygous glycogen storage disease type-Ia patients carrying the *G6PC1*-R83C variant.

Glycogen storage disease type-Ia (GSD-Ia; MIM232200), also known as von Gierke disease, is an autosomal recessive metabolic disorder caused by a deficiency in glucose-6-phosphatase-α (G6Pase-α or G6PC1) that is expressed primarily in the liver, kidney cortex, and intestine[1–3]. G6Pase-α catalyzes the hydrolysis of glucose-6-phosphate (G6P) to glucose, which is released into the blood to maintain euglycemia. A prevalent pathogenic variant identified in Caucasian GSD-Ia patients is *G6PC1*-c.247C > T/p.R83C (hereafter referred to as *G6PC1*-R83C), which accounts for 32% of sequenced alleles in patients, and results in severe hypoglycemia due to lack of G6Pase-α enzyme activity[4]. Consequently, GSD-Ia patients, including individuals who harbor the *G6PC1*-R83C variant, manifest a metabolic phenotype of impaired glucose homeostasis, characterized by fasting hypoglycemia, hepatomegaly, nephromegaly, hyperlipidemia, hyperuricemia, lactic acidemia, and growth retardation[1–3]. Current dietary therapies[5–7] can enable GSD-Ia patients to maintain a normalized metabolic phenotype but require strict adherence, making it challenging for patients to maintain glycemic control. In addition, dietary therapies do not address the underlying pathological processes and long-term complications, hepatocellular adenoma/carcinoma (HCA/HCC) and renal disease still occur in metabolically compensated patients[1–3,8]. Gene therapies that address liver G6Pase-α deficiency offer an opportunity

[1]Section on Cellular Differentiation, Division of Translational Medicine, Eunice Kennedy Shriver National Institute of Child Health and Human Development, National Institutes of Health, Bethesda, MD 20892, USA. [2]BEAM Therapeutics, Cambridge, MA 02142, USA. ✉e-mail: chouja@mail.nih.gov

to reduce disease burden and improve the quality of life for GSD-Ia patients.

A phase III clinical trial for GSD-Ia (NCT05139316) using a recombinant adeno-associated virus (rAAV) vector[9–12] to deliver a normal *G6PC1* gene is currently in progress. While a promising therapy, there is insufficient data to understand if lifelong episomal transgene expression can be maintained at a therapeutic level in humans[13,14], and many patients have pre-existing antibodies to the rAAV vector[15,16], making them ineligible for treatment. Conversely, therapeutic gene editing strategies that correct specific disease-causing mutations within the endogenous *G6PC1* gene are anticipated to be durable. Two CRISPR/Cas9-based gene editing strategies have been demonstrated in vivo to correct the GSD-Ia phenotype in a homozygous knock-in mouse model, *G6pc*-R83C[17,18]. One was a homology directed repair (HDR)[19,20] strategy utilizing AAV to deliver the editing reagents to neonatal *G6pc*-R83C mice[17]. The second was a non-homologous end joining (NHEJ)[19,20] strategy utilizing lipid nanoparticles (LNP)[21,22] to deliver the editing DNA oligonucleotides to 2- to 4-week-old *G6pc*-R83C mice[18]. Both studies showed that the edited *G6pc*-R83C mice survived to adulthood without suffering hypoglycemic seizures. However, the efficacy of the HDR-mediated editing in non-dividing cells was low[17] and the success of the NHEJ-mediated editing was dependent on the efficiency of inserting the editing DNA oligonucleotide in the correct orientation[18].

In this study, we examine the efficacy of base editing, a CRISPR-based gene editing approach that installs single base substitutions, using a newly created humanized knock-in *G6PC1*-R83C mouse strain (huR83C) (Supplementary Fig. 1). This model was generated by insertion of the entire human *G6PC1*-R83C (*G6PC1*-c.247C > T) coding sequence into exon 1 of the mouse *G6pc* gene, in a manner that knocked out the native mouse *G6pc* gene expression while placing the expression of the human *G6PC1*-R83C under the control of the mouse *G6pc* promoter/enhancer. We show that the huR83C mice lacked hepatic and renal G6Pase-α activity and manifested a human GSD-Ia pathophysiology, similar to the global *G6pc*-deficient (*G6pc*-/-) mice[23].

The prevalent pathogenic *G6PC1*-c.247C > T/p.R83C variant contains a single G > A transition mutation on the complementary strand that can be targeted by the adenine base editors (ABEs) that enable the programmable conversion of A•T to G•C in genomic DNA, and in principle could be used to precisely correct this mutation[24–29]. In this study, we evaluate the efficacy of a newly-engineered ABE to correct metabolic defects associated with GSD-Ia in a humanized mouse model. Messenger RNA encoding the ABE and guide RNA targeting the *G6PC1*-R83C variant were encapsulated in a LNP[21,22] formulation hereafter referred to as BEAM-301 and delivered via a single systemic administration either in newborn (NB) mice within 48 hours post birth or at 3 weeks (3W) of age. BEAM-301 at both a high dose (301H, 1.5 mg/kg) and a low dose (301L, 0.75 mg/kg) were evaluated, and the outcomes monitored to 53 weeks of age. We show that the efficiency of correcting the *G6PC1*-R83C variant correlates with the activity of hepatic G6Pase-α restored in the edited mice. Moreover, the edited mice harboring ≥3 units of hepatic G6Pase-α activity displayed an improved metabolic profile, sustained 24 hours of fasting, and survived long-term (through 1 year). In contrast, untreated littermates expressed negligible hepatic G6Pase-α activity and exhibited fasting hypoglycemia, with a 3-week survival rate of 39%. Significantly, no decline in hepatic G6Pase-α activity was observed over the 53-week study. In summary, we have developed a base-editing strategy for in vivo correction of a pathogenic *G6PC1* variant in the native genetic locus, that may offer a durable and highly efficient therapeutic for patients with GSD-Ia.

## Results

### Generation of huR83C, a humanized *G6PC1*-R83C mouse model for GSD-Ia

Mammalian G6Pase-α proteins share 87–91% amino acid sequence identity and are functionally equivalent[1–3]. We have shown that human and mouse G6Pase-α have identical structure-function relationships, and that the *G6pc*-R83C variant is pathogenic in mice[17]. In this study, we used wild-type (mR83) mice carrying two functional mouse (m) *G6pc*-R83 alleles, the homozygous (huR83C) mice carrying two alleles with the human (hu) *G6PC1*-R83C variant knocked-in at the *G6pc* locus, and the heterozygous (mR83/huR83C) mice. GSD-Ia is an autosomal recessive disorder[1–3], and, as expected, the mR83 and mR83/huR83C littermates display indistinguishable wild-type phenotypes and, therefore, were used as controls.

Liver microsomal G6Pase-α activity in 3-week-old wild-type, heterozygous, and homozygous huR83C mice averaged 330.4 ± 36.3, 176.2 ± 13.7, and 2.2 ± 0.3 units, respectively. Kidney microsomal G6Pase-α enzyme activity averaged 607.2 ± 49.2, 267.8 ± 18.2, and 2.6 ± 0.4 units, across respective cohorts. The lower level of quantitation for the microsomal G6Pase-α assay is 2 units, indicating that the huR83C mice have null hepatic and renal *G6pc* loci. As expected, the huR83C mice manifest a GSD-Ia phenotype of impaired glucose homeostasis[1–3]. Consistent with this, of the 42 untreated huR83C mice studied, only 16 (39%) survived to age 3 weeks.

### Base editing strategy for correction of *G6PC1*-R83C

Genetic correction of the pathogenic human *G6PC1*-R83C (*G6PC1*-c.247C > T) variant requires the conversion of an A:T base pair to a G:C base pair. To target this allele, a guide RNA was designed with an saCas9 'NNGRRT' protospacer-adjacent motif (PAM), with target adenine (noted at position 16 in Fig. 1a). The base-editor used harbors an saCas9 nickase domain and a TadA deaminase domain engineered to maximize editing of the target adenine and reduce editing of bystander adenines[26]. Base-editing for correction of the *G6PC1*-R83C allele was first confirmed in vitro using primary human hepatocytes (Supplementary Fig. 2). Subsequently, the guide RNA and messenger RNA encoding the base-editor were co-formulated into an LNP to generate BEAM-301, an investigational base-editing therapy for GSD-Ia patients who harbor the *G6PC1*-R83C variant. Editing at the target adenine corrects the *c.247C > T*/p.R83C variant back to wild-type *c.247C*/p.R83. There are two neighboring bystander adenine bases at positions 18 and 22 (Fig. 1a). An edit at position 18 introduces a synonymous variant while an edit at position 22 introduces a nonsynonymous p.Y85H variant.

Given the high rate of neonatal lethality in the homozygous huR83C mice, we first validated that BEAM-301 administration can correct the *G6PC1*-R83C (*G6PC1*-c.247C > T) allele in the livers of 10–12-week-old heterozygous (mR83/huR83C) mice (Fig. 1). BEAM-301 was dosed systemically via tail vein injection of 0.1 to 3.0 mg/kg and the base-editing efficiency of the *G6PC1* alleles was analyzed via next-generation sequencing (NGS) in isolated liver slices at 7 days post-dose (Fig. 1b). The editing efficiency for the conversion of the GSD-Ia *G6PC1*-R83C variant to wild-type *G6PC1*-R83 (*G6PC1*-c.247C) correlated positively with administered dose level of BEAM-301. A low frequency of both bystander edits was also detected. The functional impact of the non-synonymous Y85H edit (position 22) was evaluated using transient expression assays in COS1 cells[30]. The G6Pase-α-Y85H variant lacked detectable phosphohydrolase activity (Supplementary Fig. 3), classifying it as a pathogenic G6Pase-α variant.

The editing efficiency of BEAM-301, administered to heterozygous mice at 1.5 mg/kg (301H) is shown in Fig. 1c. An average of 40.3% of sequencing reads encoded wild-type G6Pase-α, whereas 5.5% contained the nonsynonymous bystander edit resulting in the Y85H allele, 0.3% contained only a synonymous bystander edit, and 0.7% exhibited indels. Since the predominant editing outcome is correction to wild-type *G6PC1*-R83, and all other outcomes are expected to yield inactive G6Pase-α, similar to the pre-existing *G6PC1*-R83C variant, the editing induced by BEAM-301 is expected to provide an overall functional benefit.

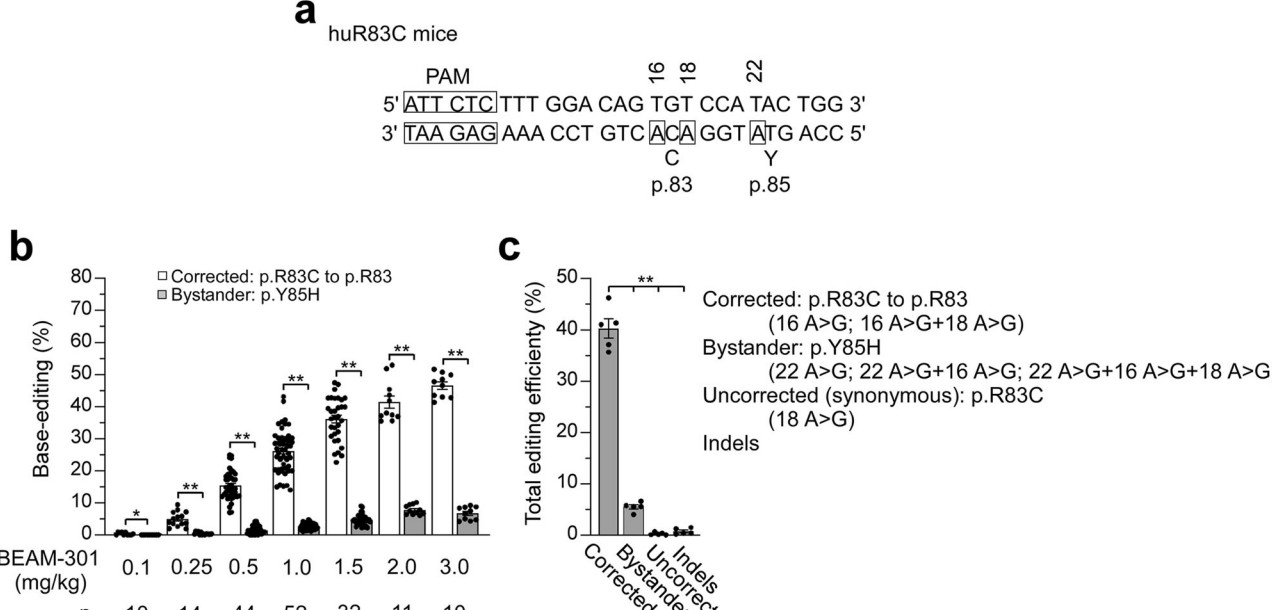

**Fig. 1 | In vivo correction of the *G6PC1*-R83C allele by base editing.** The 10–12-week-old heterozygous (mR83/huR83C) mice, not fasted, were treated by systemic administration of BEAM-301 and the target site sequence was analyzed one week later via next-generation sequencing (NGS). **a** Target site sequence for p.R83C to p.R83 correction, the p.R83 synonymous variant and the pathogenic p.Y85H bystander variant. Nucleotides on the complementary strand edited by BEAM-301 are bracketed. **b** Correlation of dose levels of BEAM-301 to the editing efficiency for the conversion of the p.R83C (*G6PC1-c.247C > T*) variant to wild-type p.R83 (*G6PC1-c.247C*) along with the p.Y85H mutation. The *n* numbers for each dosage of BEAM-301 in mg/kg are listed. **c** Allele frequencies resulting from dosing heterozygous mice at 1.5 mg/kg (301H): p.R83C to p.R83 (correction); p.Y85H bystander mutation; Uncorrected (synonymous) editing (R83C); and Indels (*n* = 5). Statistics were performed using a two-tailed unpaired *T* test. Data are presented as Mean values ± SEM, and individual data points for each animal are displayed. * denotes *p* < 0.05, ** denotes *p* value < 0.005.

## Off-target analysis

Validation of off-target sites was conducted in primary human hepatocytes, the primary target for base editing. The potential off-target editing sites were identified using a 2-stage approach: nomination of candidate off-target sites and validation of off-target sites in treated cells.

Potential off-target sites were nominated using two complementary biochemical assays: oligonucleotide enrichment and sequencing (ONE-seq) and Digenome-seq. ONE-seq leverages in silico predictions to construct a library of potential off-target sites for high-sensitivity detection of base deamination and nicking events; Digenome-seq detects genome-wide base deamination and nicking events on naked genomic DNA. In aggregate, 591 unique candidate sites were selected for further evaluation in cells; including sites nominated from ONE-seq or Digenome-seq experiments; sites that were identified in silico as low-mismatch sites (but nominated by either assay); and others (including the 1 on-target site, and 20 negative control sites).

Primary human hepatocytes from 3 donors were treated with BEAM-301 at or near saturation of on-target editing. The rhAmpSeq method was used to amplify the candidate off-target sites in parallel in treated and untreated (control, donor-matched) cells, and after NGS processing, off-target sites were assessed for enriched editing in treated versus untreated (control, donor-matched) cells. A single site, located within an intron, was shown to have <1% editing following treatment at saturation (Supplementary Fig. 4).

## Pathophysiology of 301H-dosed huR83C mice at 3 weeks of age

Newborn (NB) huR83C mice, were treated by systemic administration of BEAM-301, at a dose level of 1.5 mg/kg (301H), and metabolic correction was assessed at 3 weeks of age. Of the eight mice treated, two mice expressed background hepatic and renal G6Pase-α activity of 2.0

and 2.2 units, respectively, and genomic analysis confirmed the absence of base-editing in their livers, suggesting failed administration of BEAM-301 rather than failure to target the locus. Therefore, they were excluded from further analysis. In the remaining six mice, liver microsomal G6Pase-α activities averaged 163.8 ± 48.5 units (Fig. 2a), statistically equivalent to heterozygous control mice which had 176.2 ± 13.7 units. Individually, hepatic G6Pase-α activity ranged from 33 to 289 units (Fig. 2a) and revealed a positive correlation with base-editing efficiency (R83C > R83), with a maximum rate of ~60% (Fig. 2b). Published data show that restoring ≥5 units (3%) of hepatic G6Pase-α activity in GSD-Ia mice enables them to maintain blood glucose homeostasis[9–11,17,18]. As expected, at age 3 weeks and in the fed state, blood glucose levels in 301H-treated huR83C mice were similar to their control littermates (Fig. 2c). Additionally, the 301H-treated huR83C mice were able to survive 24 hours of fasting challenges, with fasting blood glucose levels indistinguishable from the control mice (Fig. 2c). In contrast, the untreated huR83C mice exhibited marked hypoglycemia within 60 to 75 min of fasting (Supplementary Fig. 5), a hallmark of GSD-Ia[1–3] as was previously shown for the untreated *G6pc-/-* mice[10].

Studies have shown that G6Pase-α activity in wild-type (mR83) or heterozygous (mR83/huR83C) mice is expressed in hepatocytes and kidney cortex[1–3]. Consistently, histochemical enzyme analysis showed that in the heterozygous (mR83/huR83C) mouse, hepatic G6Pase-α activity was distributed throughout the hepatocytes, with higher levels in proximity to the blood vessels, while in the mR83/huR83C mouse kidney, G6Pase-α activity was restricted to the cortex (Fig. 2d). As expected, G6Pase-α activity was not detectable in the liver or kidney sections of the huR83C mice (Fig. 2d). In the NB-301H-dosed mice expressing wild-type levels of hepatic G6Pase-α activity, the distribution pattern was similar to that of the mR83/huR83C control mice (Fig. 2d). The edited mice expressing lower levels of hepatic G6Pase-α activity had a focal pattern of expression with G6Pase-α enzymatic

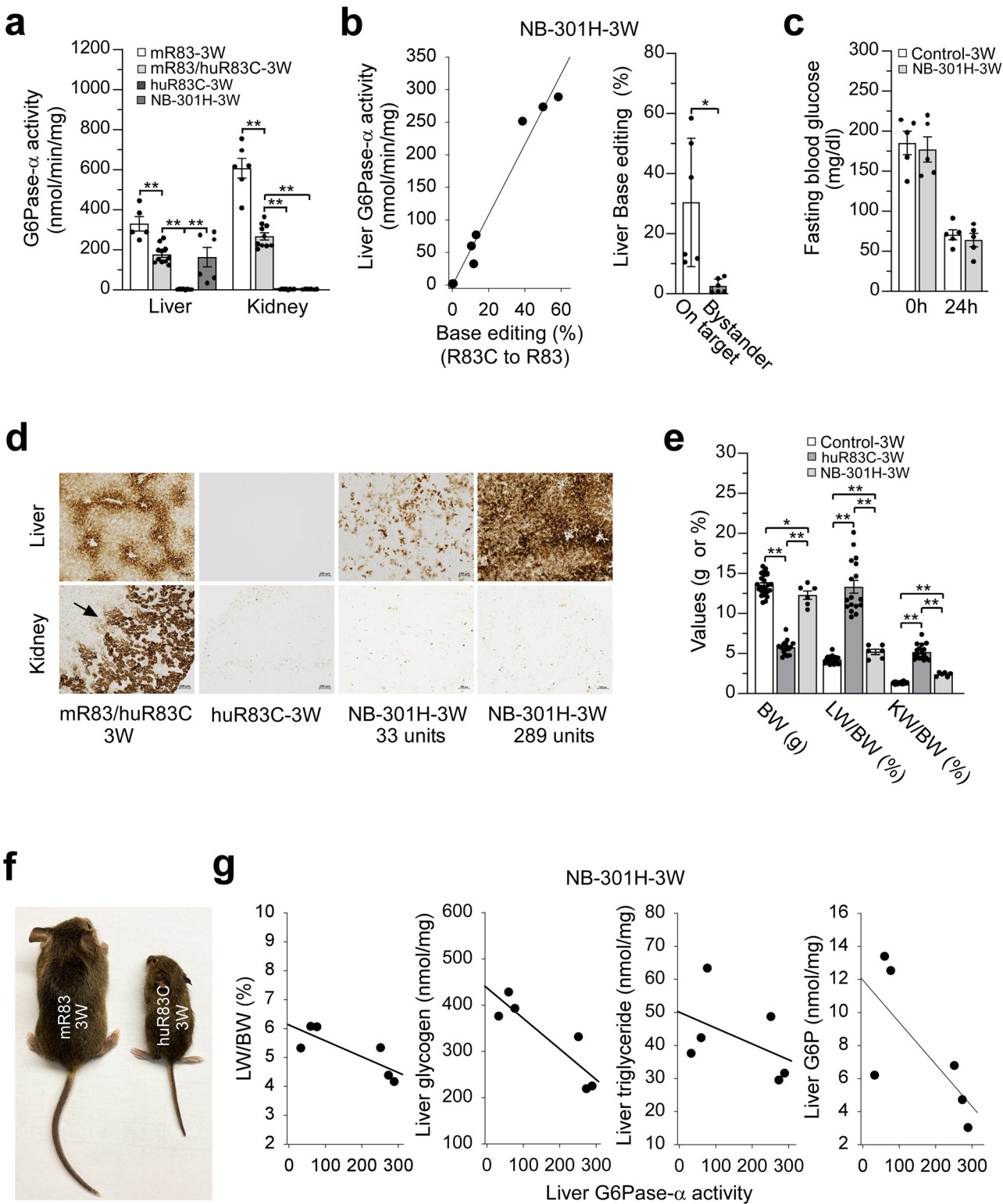

activity associated with the vasculature (Fig. 2d). These findings are consistent with those observed in rAAV-mediated gene augmentation in *G6pc-/-* mice[10]. Consistent with the enzyme activity assays, no histochemical staining of G6Pase-α was detected in the kidneys of the edited huR83C mice (Fig. 2d).

The untreated huR83C mice manifested a typical GSD-Ia phenotype of hepatomegaly, nephromegaly, and growth retardation (Fig. 2e, f). At 3 weeks of age, the body weight (BW) of huR83C mice measured

significantly lower than their sex-matched control littermates (Fig. 2e, f) and their liver weight (LW)/BW and kidney weight (KW)/BW ratios were $13.3 \pm 0.8\%$ and $5.2 \pm 0.2\%$, respectively, which were significantly higher than the respective ratios in the control mice of $4.2 \pm 0.1\%$ and $1.30 \pm 0.02\%$ (Fig. 2e). In the NB-dosed huR83C mice, the BW, LW/BW, and KW/BW values closely approximate those of the control littermates (Fig. 2e). Despite renal G6Pase-α activity being non-detectable in both untreated and NB-dosed huR83C mice at 3 weeks of age,

**Fig. 2 | Phenotype of 3-week-old untreated and NB-301H-dosed huR83C mice.** Newborn (NB) huR83C mice, non-fasted, were treated with a high dose of BEAM-301 (301H) at 1.5 mg/kg. At age 3 weeks, the phenotype of the resulting NB-301H mice (NB-301H-3W) was evaluated and compared to age-matched unedited huR83C (huR83C-3W), wild-type (mR83-3W), and heterozygote (mR83/huR83C-3W) mice. **a** Liver and kidney microsomal G6Pase-α activity. Liver (mR83-3W, *n* = 5; mR83/huR83C-3W, *n* = 11; huR83C-3W, *n* = 8; NB-301H-3W, *n* = 6) and kidney (mR83-3W, *n* = 6; mR83/huR83C-3W, *n* = 10; huR83C-3W, *n* = 8; NB-301H-3W, *n* = 6). **b** Restoration of hepatic G6Pase-α activity as a function of base editing efficiency in NB-301H-3W mice (*n* = 6) along with on-target and bystander values of liver base editing. **c** Fasting blood glucose levels in control (mR83-3W and mR83/huR83C-3W; *n* = 5) and NB-301H-3W (*n* = 5) mice. **d** Histochemical analysis of liver and kidney

G6Pase-α activity in control (mR83-3W and mR83/huR83C-3W, *n* = 6), untreated (huR83C-3W, *n* = 6), and NB-301H-3W (*n* = 6) mice. Each image represents an individual mouse. The arrow indicates the kidney cortex. Scale bar = 100 μm. The numbers represent hepatic G6Pase-α activity expressed in the mice. **e** Body weight (BW), liver weight (LW)/BW, and kidney weight (KW)/BW values of control (mR83-3W and mR83/huR83C-3W, *n* = 25), untreated (huR83C-3W, *n* = 17), and NB-301H-3W (*n* = 6) mice. **f** Size comparison of mR83 and huR83C mice, showing growth retardation of the huR83C mice at age 3 weeks. **g** Restoration of hepatic G6Pase-α activity as a function of LW/BW values and hepatic levels of glycogen, triglyceride, and G6P in the NB-301H-3W mice (*n* = 6). Statistics were performed using a two-tailed unpaired *T* test. Data are presented as Mean values ± SEM, and individual data points for each animal are displayed. * denotes *p* < 0.05, ** denotes *p* value < 0.005.

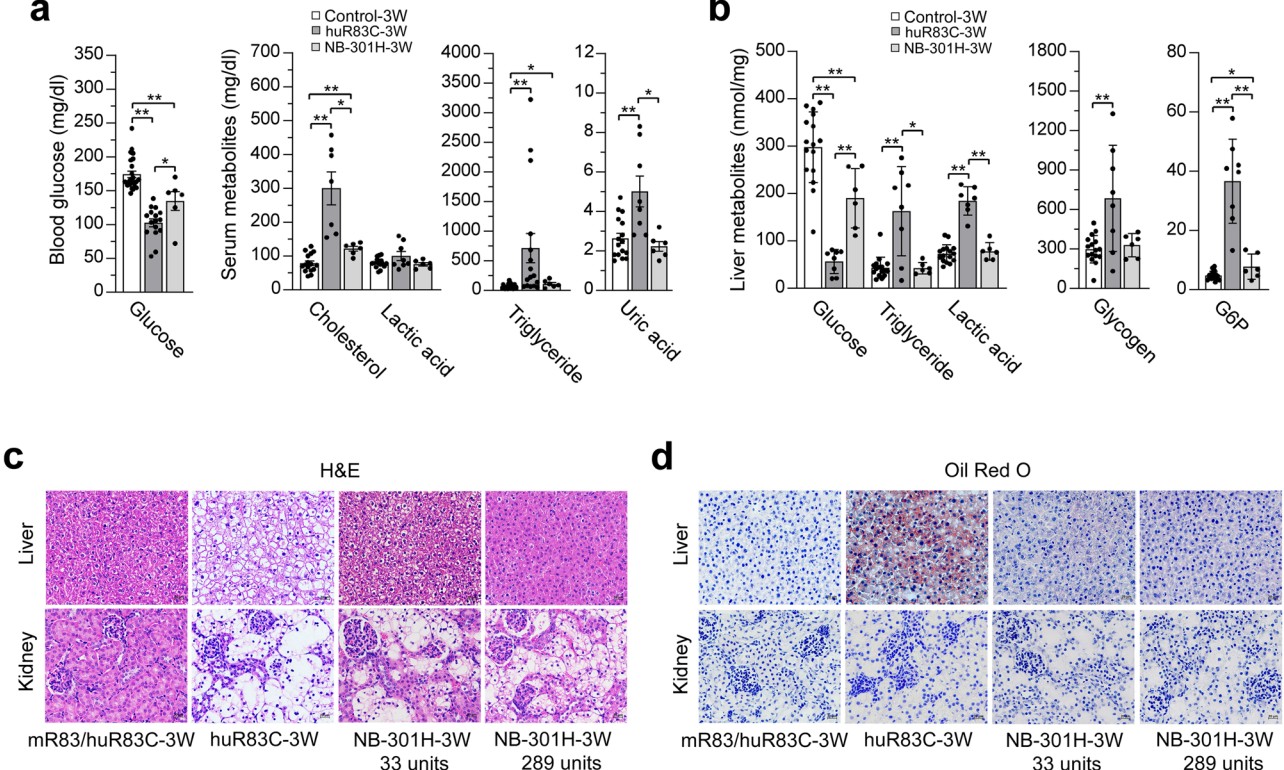

**Fig. 3 | Biochemistry of 3-week-old untreated and NB-301H-dosed huR83C mice.** Newborn (NB) huR83C mice, non-fasted, were treated with a high dose of BEAM-301 (301H) at 1.5 mg/kg. At age 3 weeks, biochemical phenotype of the edited mice was analyzed and compared to the age matched unedited huR83C. The 3-week-old mR83 and mR83/huR83C littermates displaying a wild-type phenotype were used as the controls. **a** Blood glucose levels (control-3W, *n* = 24; huR83C-3W, *n* = 16; NB-301H-3W, *n* = 6), serum cholesterol levels (control-3W, *n* = 16; huR83C-3W, *n* = 7; NB-301H-3W, *n* = 6), serum triglyceride levels (control-3W, *n* = 24; huR83C-3W, *n* = 16; NB-301H-3W, *n* = 6), serum lactate and uric acid levels (control-3W, *n* = 16; huR83C-3W, *n* = 8; NB-301H-3W, *n* = 6). **b** Liver glucose, triglyceride, lactate, glycogen, and G6P levels in control (*n* = 16), huR83C-3W (*n* = 8), and NB-301H-3W (*n* = 6) mice. **c** Hematoxylin and eosin (H&E)-staining of liver and kidney sections in

mR83/huR83C-3W (*n* = 6), huR83C-3W (*n* = 6), and NB-301H-3W (*n* = 6) mice. The liver and kidney in all experimental animals were examined. A single image from an individual mouse, representative of the results in all mice, is shown to illustrate the results. Scale bar = 20 μm. The numbers represent hepatic G6Pase-α activity expressed in the mice. **d** Oil Red O staining of liver and kidney sections in mR83/huR83C-3W (*n* = 6), huR83C-3W (*n* = 6), and NB-301H-3W (*n* = 6) mice. The liver and kidney in all experimental animals were examined. A single image from an individual mouse, representative of the results in all mice, is shown to illustrate the results. Scale bar = 20 μm. The numbers represent hepatic G6Pase-α activity expressed in the mice. Statistics were performed using a two-tailed unpaired *T* test. Data are presented as Mean values ± SEM, and individual data points for each animal are displayed. * denotes *p* < 0.05, ** denotes *p* value < 0.005.

restoration of liver G6Pase-α activity by base-editing benefitted the kidney, with the KW/BW value decreasing from 5.2 ± 0.2% for the untreated huR83C mice to 2.4 ± 0.1% for the edited mice, compared to 1.30 ± 0.02% for the control mice (Fig. 2e). In the 3-week-old 301H-dosed mice, the LW/BW values were inversely correlated with the level of hepatic G6Pase-α activity restored (Fig. 2g), demonstrating a direct relationship between liver G6Pase-α activity and hepatomegaly. Hepatomegaly in GSD-Ia is caused by elevated accumulation of glycogen/neutral fat[1–3]. Notably, in the NB-dosed mice, hepatic levels of

both glycogen and triglyceride were inversely correlated with the level of restored hepatic G6Pase-α activity (Fig. 2g). In the NB-dosed mice, hepatic G6P levels were also inversely correlated with hepatic G6Pase-α activity restored (Fig. 2g).

The untreated huR83C mice also manifested hypoglycemia, hypertriglyceridemia, hypercholesterolemia, and hyperuricemia (Fig. 3a), characteristic of GSD-Ia[1–3]. Serum lactate concentrations were not significantly elevated in 3-week-old, untreated huR83C mice (Fig. 3a), consistent with previous observations in the *G6pc-/-* mice[23].

Compared to untreated huR83C mice, the NB-dosed mice displayed improved blood/serum metabolite profiles (Fig. 3a). A deficiency in G6Pase-α reduces hepatic glucose production and reprograms G6P metabolism, leading to increased G6P accumulation, glycogen synthesis, and glycolysis[31]. Compared to the controls, hepatic glucose levels in the 3-week-old untreated huR83C mice were markedly reduced along with increased hepatic levels of triglyceride, glycogen, lactate, and G6P (Fig. 3b). The NB-dosed mice displayed normal hepatic levels of triglyceride, lactate, and glycogen, although hepatic glucose levels remained reduced and hepatic G6P levels remained elevated, compared to the controls (Fig. 3b).

Hematoxylin and eosin (H&E) staining showed that the 3-week-old untreated huR83C mice displayed no histological abnormalities in the liver except for a diffuse mosaic pattern of hepatocytes, consistent with their increased glycogen accumulation (Fig. 3c), as seen in GSD-Ia patients. H&E staining also showed that the 3-week-old untreated huR83C mice displayed marked glycogen accumulation in the renal tubular epithelial cells, resulting in enlargement and compression of the glomeruli (Fig. 3c). Oil Red O staining confirmed the increases in neutral fat accumulation in the liver with little or no fat accumulation in the kidney of 3-week-old huR83C mice (Fig. 3d), consistent with GSD-Ia and the *G6pc*-R83C mice[17].

Compared to 3-week-old untreated huR83C mice, H&E and Oil Red O staining confirmed the marked reduction in hepatic levels of glycogen and neutral fat in the NB-dosed mice that inversely correlated to the extent of hepatic G6Pase-α activity restored (Fig. 3c, d). Little to no fat accumulation was observed in the edited kidneys (Fig. 3d). In summary, the huR83C mice manifested a phenotype of impaired glucose homeostasis that mimics human GSD-Ia, validating a humanized knock-in mouse model for GSD-Ia. Moreover, the NB-dosed huR83C mice displayed a markedly improved metabolic phenotype, establishing the efficacy of base-editing in correcting metabolic abnormalities in GSD-Ia mice.

## Pathophysiology of 301H-dosed huR83C mice at 8 weeks of age
In contrast to the tight range of editing rates observed in BEAM-301-treated adult heterozygous (mR83/huR83C) mice (Fig. 1), homozygous (huR83C) mice edited at birth exhibited a large variation in both the editing efficiency and the hepatic G6Pase-α activity restored (Fig. 2).

To further investigate this phenomenon, we infused 301H into NB and non-fasted 3W huR83C mice and compared their phenotypes at age 8 weeks. Liver microsomal G6Pase-α activity in 8-week-old wild-type and heterozygous mice measured $258.9 \pm 14.3$ and $174.7 \pm 7.0$ units on average, respectively (Fig. 4a). At age 8 weeks, liver microsomal G6Pase-α activity in 3W-dosed huR83C mice averaged $150.8 \pm 27.4$ units, over 2-fold higher than NB-dosed mice that averaged $65.0 \pm 22.8$ units (Fig. 4a). Liver microsomal G6Pase-α activities in the NB-dosed mice varied 35-fold, from 5 to 173 units, with a corresponding range in base-editing efficiency of 0.8% to 35% as analyzed via NGS. In contrast, liver microsomal G6Pase-α activities in the 3W-dosed mice varied only 10-fold, from 30 to 294 units, with a range in base-editing efficiency of 2.9% to 36.6%. Based on these data, dosing huR83C mice with 301H at 3 weeks of age appears to reduce editing variability and increase editing efficiency. Again, restoration of hepatic G6Pase-α activity correlated with the editing efficiency for all cohorts dosed, irrespective of age at dosing (Fig. 4b). Importantly, all NB ($n = 8$) and 3W ($n = 9$) 301H-dosed huR83C mice survived to the terminal timepoint of the study (8 weeks) and could sustain 24 hours of fasting (Fig. 4c).

In the kidney of huR83C mice, systemic infusion of 301H restored little or no G6Pase-α expression with kidney microsomal G6Pase-α activity in NB-dosed ($4.5 \pm 0.44$ units) and 3W-dosed ($3.3 \pm 0.51$ units) much lower than that in the 8-week-old wild-types ($520.8 \pm 55.5$) and heterozygotes ($270.2 \pm 18.8$ units) (Fig. 4a).

At age 8-weeks, the BW values of 3W-301H-dosed huR83C mice were lower than both their sex-matched littermate controls and NB-301H-dosed huR83C mice (Fig. 4d), which can be explained by the significantly lower baseline BW values of the 3-week-old huR83C mice prior to dosing (Fig. 2e, f). The 8-week-old 301H-dosed huR83C mice continued to display hepatomegaly and nephromegaly (Fig. 4d) and the LW/BW values and hepatic levels of glycogen and G6P were inversely correlated with the level of hepatic G6Pase-α activity restored (Fig. 4e).

Importantly, all 301H-dosed huR83C mice displayed normal blood/serum levels of glucose, cholesterol, triglyceride, lactic acid, and uric acid (Fig. 4f). Hepatic levels of triglyceride in all edited huR83C mice were similar to those of littermate controls but hepatic levels of glycogen and G6P were elevated (Fig. 4g). Notably, at age 8 weeks, hepatic levels of glucose and lactate were completely normalized in the 3W-dosed but not in the NB-dosed huR83C mice (Fig. 4g).

## Pathophysiology of 301L-dosed huR83C mice at 8 weeks of age
The efficacy of reducing the dose of editing reagents was assessed by repeating the earlier experiment at half the original dose of BEAM-301, namely 0.75 mg/kg (301 L) and monitoring the restoration of metabolic parameters at 8 weeks of age. Eighteen huR83C mice were treated as newborns and provided with access to standard mouse chow, but 9 died prematurely, resulting in an 8-week survival rate of 50%. Among the 21 huR83C mice treated in a similar manner but at age 3 weeks, 3 died prematurely, resulting in an 8-week survival rate of 86%. Due to the broad range in hepatic G6Pase-α restored in NB-dosed mice noted above, it is likely that 50% of the low-dose, NB-treated huR83C mice failed to express sufficient hepatic G6Pase-α activity to support their survival.

The liver microsomal G6Pase-α activity in 8-week-old control mice, the surviving NB-dosed ($n = 9$) and 3W-dosed ($n = 8$) huR83C mice averaged $238.9 \pm 17.6$, $24.5 \pm 5.1$, and $31.3 \pm 7.3$ units, respectively (Fig. 5a). Again, hepatic G6Pase-α activity restored in the edited mice correlated with editing efficiency (Fig. 5b). Moreover, all 8-week-old edited huR83C mice could sustain 24 hours of fasting (Fig. 5c).

As was seen with the high dose, the low dose (301 L) 3W-dosed huR83C mice had significantly lower BW values than their littermate controls and were also lower than the NB-treated huR83C mice, most likely due to the lower baseline weight prior to dosing at 3W (Fig. 5d). Both the NB and 3W 301L-dosed cohorts continued to manifest hepatomegaly and nephromegaly (Fig. 5d). The LW/BW values and hepatic levels of glycogen and G6P of the 8-week-old 301L-dosed mice were inversely correlated with the level of hepatic G6Pase-α activity restored (Fig. 5e), as expected by the direct relationship between liver G6Pase-α activity and metabolic control. While the NB-dosed huR83C mice displayed increased levels of serum cholesterol (Fig. 5f), the 3W-dosed mice displayed a normal blood/serum metabolite profile at age 8 weeks (Fig. 5f), reflecting the lower hepatic G6Pase-α activity restored in the NB-dosed mice compared to the 3W-dosed mice. Compared to the 8-week-old littermate controls, both NB- and 3W-dosed huR83C mice displayed normal hepatic triglyceride levels, although their hepatic levels of glucose remained reduced and hepatic levels of glycogen, G6P, and lactate remained elevated (Fig. 5g).

## Long-term correction of metabolic abnormalities of the huR83C mice
To evaluate the persistence and therapeutic potential of in vivo BEAM-301-mediated base editing, we undertook a study up to 53 weeks of age in which NB huR83C mice were infused with a high dose (301H) and 3W huR83C mice were infused with a 50% lower dose (301 L) of the BEAM-301 editing reagents. Although the 53-week study was not designed to establish the threshold of hepatic G6Pase-α activity required for tumor

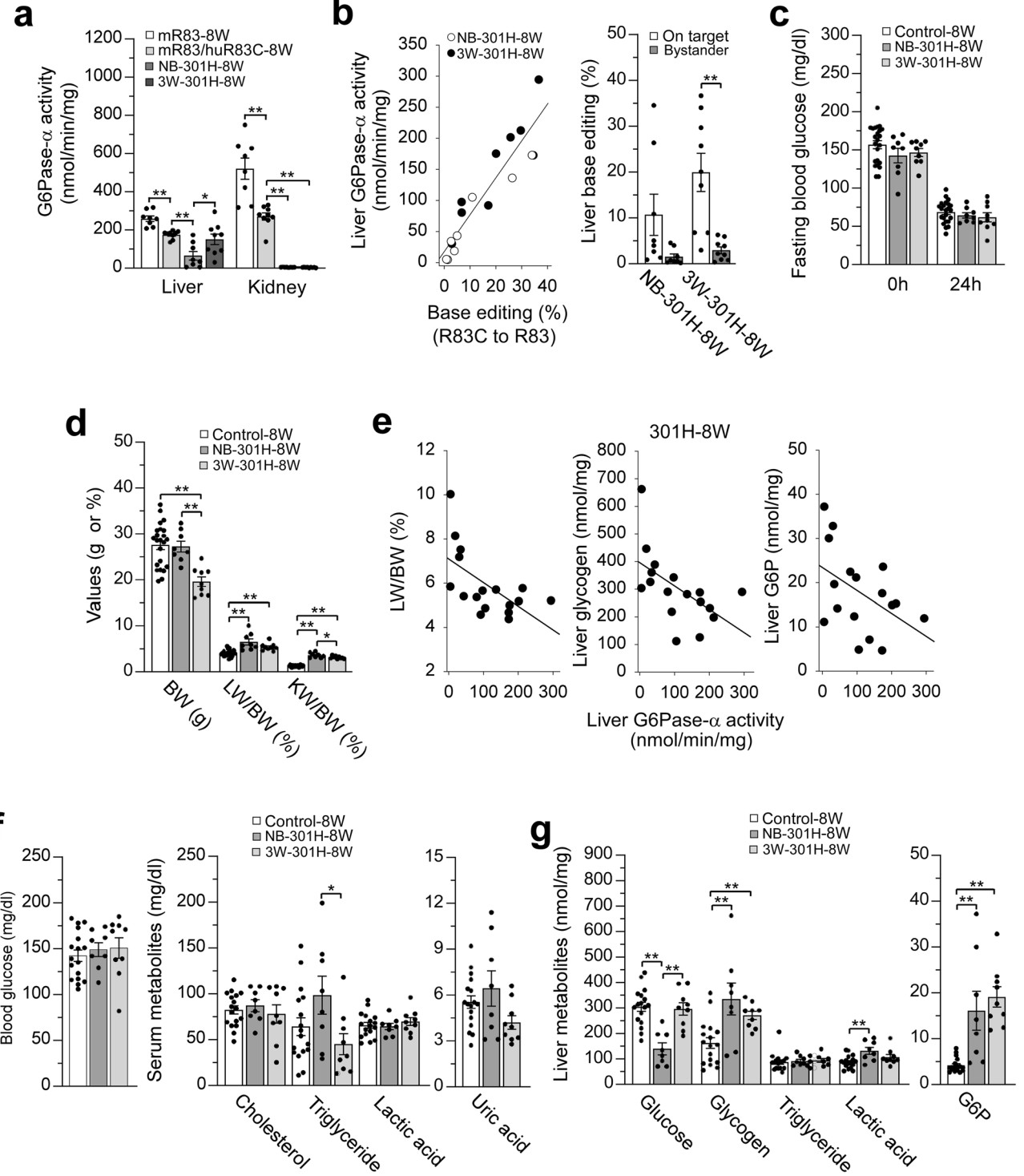

**Fig. 4 | Phenotypic analysis of 301H-dosed huR83C mice at age 8 weeks.** New-born (NB) and 3-week-old (3W) huR83C mice, non-fasted, were treated with 301H (BEAM-301 at 1.5 mg/kg) and the phenotype of the resulting NB-301H-8W and 3W-301H-8W mice was analyzed at age 8 weeks. The sex-matched mR83-8W and mR83/huR83C-8W littermates displaying a wild-type phenotype were used as the controls. **a** Liver and kidney microsomal G6Pase-α activity in mR83-8W ($n = 8$), mR83/huR83C-8W ($n = 9$), NB-301H-8W ($n = 8$), and 3W-301H-8W ($n = 9$) mice. **b** Restoration of hepatic G6Pase-α activity as a function of base editing efficiency along with on-target and bystander values of liver base editing. NB-301H-8W ($n = 8$); 3W-301H-8W ($n = 9$). **c** Fasting blood glucose levels in control ($n = 23$), NB-301H-8W ($n = 8$), and 3W-301H-8W ($n = 9$) mice. **d** BW, LW/BW, and KW/BW values of control ($n = 23$), NB-301H-8W ($n = 8$), and 3W-301H-8W ($n = 9$) mice. **e** Restoration of hepatic G6Pase-α activity as a function of LW/BW values and hepatic levels of glycogen and G6P in the 301H-8W mice ($n = 17$), including NB-301H-8W ($n = 8$) and 3W-301H-8W ($N = 9$) mice. **f** Blood glucose and serum cholesterol, triglyceride, lactate, and uric acid levels in control ($n = 17$), NB-301H-8W ($n = 8$), and 3W-301H-8W ($n = 9$) mice. **g** Liver glucose, glycogen, triglyceride, lactate, and G6P levels in control ($n = 17$), NB-301H-8W ($n = 8$), and 3W-301H-8W ($n = 9$) mice. Statistics were performed using a two-tailed unpaired $T$ test. Data are presented as Mean values ± SEM, and individual data points for each animal are displayed. * denotes $p < 0.05$, ** denotes $p$ value $< 0.005$.

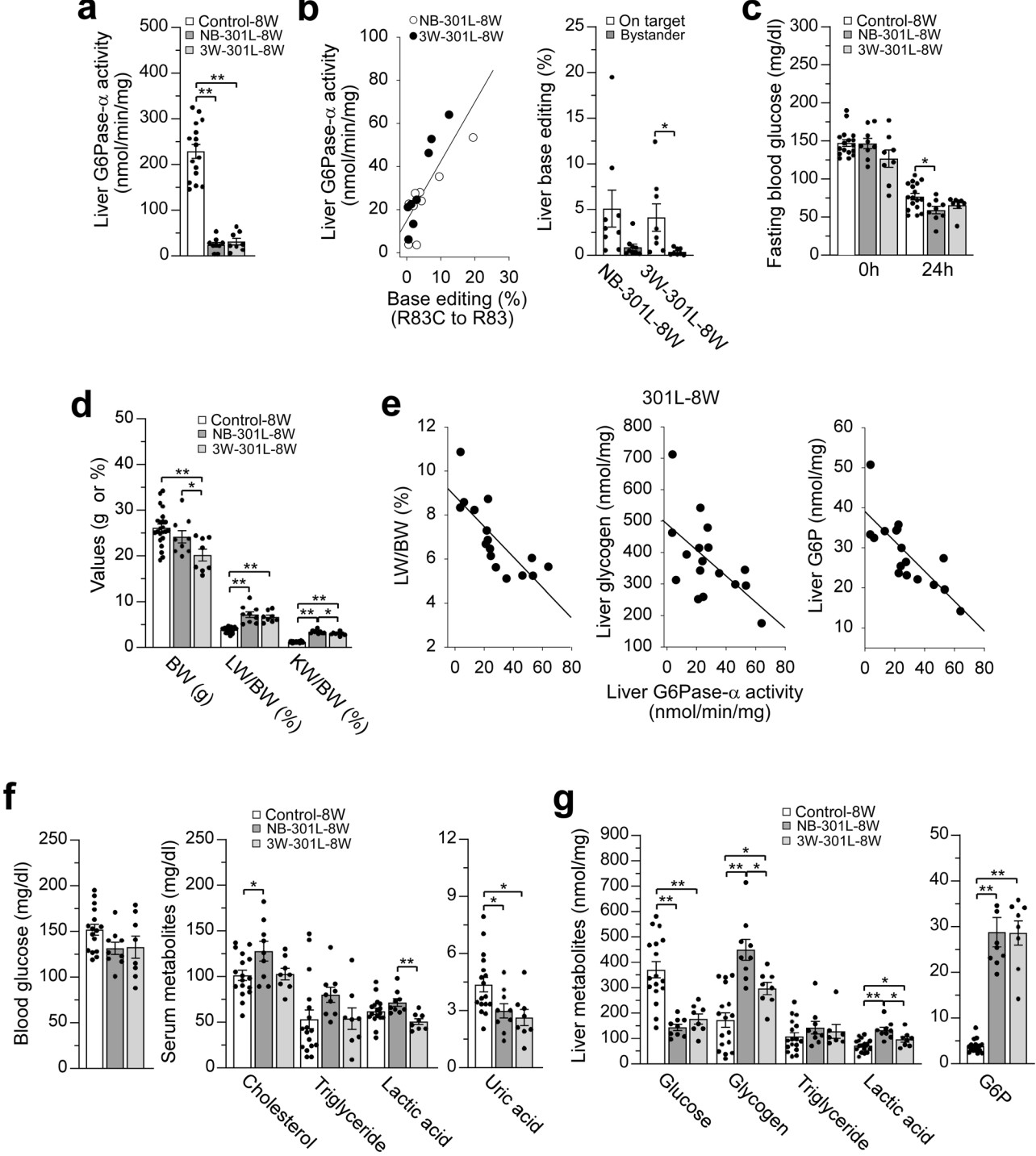

**Fig. 5 | Phenotypic analysis of 301L-dosed huR83C mice at age 8 weeks.** Newborn (NB) and 3-week-old (3W) huR83C mice, non-fasted, were treated with a low dose of BEAM-301 (301L) at 0.75 mg/kg and the phenotype of the resulting NB-301L-8W and 3W-301L-8W mice was analyzed at age 8 weeks. The sex-matched mR83 and mR83/hR83C littermates displaying a wild-type phenotype were used as the controls. **a** Liver microsomal G6Pase-α activity in control ($n = 17$), NB-301L-8W ($n = 9$), and 3W-301L-8W ($n = 8$) mice. **b** Restoration of hepatic G6Pase-α activity as a function of base editing efficiency along with on-target and bystander values of liver base editing in NB-301L-8W ($n = 9$) and 3W-301L-8W ($n = 8$) mice. **c** Fasting blood glucose levels in control ($n = 17$), NB-301L-8W ($n = 9$), and 3W-301L-8W ($n = 8$)

mice. **d** BW, LW/BW, and KW/BW values of control ($n = 23$), NB-301L-8W ($n = 9$), and 3W-301L-8W ($n = 8$) mice. **e** Restoration of hepatic G6Pase-α activity as a function of LW/BW values and hepatic levels of glycogen and G6P in the edited 301L-8W ($n = 17$) mice, including NB-301L-8W ($n = 9$) and 3W-301L-8W ($n = 8$) mice. **f** Blood glucose and serum cholesterol, triglyceride, lactate, and uric acid levels in control ($n = 17$), NB-301L-8W ($n = 9$), and 3W-301L-8W ($n = 8$) mice. **g** Liver glucose, glycogen, triglyceride, lactate, and G6P levels in control ($n = 17$), NB-301L-8W ($n = 9$), and 3W-301L-8W ($n = 8$) mice. Statistics were performed using a two-tailed unpaired $T$ test. Data are presented as Mean values ± SEM, and individual data points for each animal are displayed. * denotes $p < 0.05$, ** denotes $p$ value < 0.005.

prevention, we were interested in observing the tumor frequency between a low and high dose.

There was no premature death of the NB-301H-edited huR83C mice. Liver microsomal G6Pase-α activity in 53-week-old wild-type and NB-dosed huR83C mice averaged 211.4 ± 6.1 and 87.1 ± 19.9 units, respectively (Fig. 6a), demonstrating sustained hepatic G6Pase-α expression. Analysis of liver isolates from 53-week-old NB-dosed huR83C mice revealed base-editing rates ranging from ~0.5% to 53% and hepatic G6Pase-α activity from 2.4 to 292 units. Consistent with assessment at earlier terminal time-points, hepatic G6Pase-α activity restored in the NB-dosed mice correlated positively with base-editing efficiency (Fig. 6b). At 53 weeks of age, all NB-dosed huR83C mice ($n$ = 19) could sustain 24 hours of fasting (Fig. 6c).

At age 53 weeks, the BW and LW values of wild-type and NB-301H-dosed huR83C mice were statistically similar but the NB-dosed mice continued manifesting hepatomegaly and nephromegaly (Fig. 6d). In congruence with earlier data from younger mice, the LW/BW values, and hepatic levels of glycogen and G6P were inversely correlated with the level of hepatic G6Pase-α activity restored (Fig. 6e). The 53-week-old NB-dosed huR83C mice displayed normal levels of blood/serum metabolites, except for moderately elevated serum triglyceride levels (Fig. 6f). Compared to the wild-type controls, these NB-dosed huR83C mice displayed normal levels of hepatic glycogen, triglyceride, and lactate, although their average hepatic glucose levels remained reduced and average hepatic G6P levels remained elevated (Fig. 6g). Histological analysis showed that none of the NB-301H-dosed huR83C mice developed hepatic tumors.

Among the 21 huR83C mice treated with 301 L at age 3 weeks, 3 mice died before age 8 weeks, an additional 8 mice underwent scheduled euthanasia at 8 weeks of age, and 1 more died at age 39 weeks, resulting in a 53-week survival rate of 69%. Liver microsomal G6Pase-α activity in 53-week-old control (mR83/huR83C; $n$ = 14) and 3W-301L-dosed huR83C ($n$ = 9) mice averaged 180.1 ± 13.5 and 53.4 ± 17.6 units, respectively (Fig. 7a). The range of hepatic G6Pase-α activity in the 3W-301L-dosed huR83C mice was 2.6 to 179 units, with positive correlation to base-editing, that ranged from a rate of 0.12% to 36% (Fig. 7b). Except for the dosed huR83C mouse expressing 2.6 units of hepatic G6Pase-α activity, the other 8 edited mice could sustain 24 hours of fasting (Fig. 7c).

At age 53-weeks, the BW values of 3W-301L-dosed huR83C mice were significantly lower than their age-matched control mice (Fig. 7d). Compared to 53-week-old control mice, the 3W-301L-dosed huR83C mice continued manifesting hepatomegaly and nephromegaly (Fig. 7d), although their LW values were statistically similar to the control mice. Again, the LW/BW and hepatic levels of glycogen and G6P were inversely correlated with hepatic G6Pase-α activity restored (Fig. 7e).

Compared to the controls, the 53-week-old 3W-301L-dosed huR83C mice displayed normal levels of blood/serum metabolites (Fig. 7f) and normal levels of hepatic glycogen and lactate, although their hepatic glucose levels remained reduced and their hepatic G6P levels remained elevated (Fig. 7g). Interestingly, hepatic triglyceride levels in the 53-week-old 301L-dosed huR83C mice were markedly lower than their sex-matched littermate controls (Fig. 7g).

We have previously shown that the rAAV-G6PC1-treated $G6pc$-/- mice are leaner and protected against age-related insulin resistance[32]. At age 53 weeks, the BMI values of the 3W-301L-dosed huR83C mice were significantly lower than that of the sex-matched littermate controls (Fig. 8a), confirming that the edited mice were leaner. For insulin tolerance test, a reduced insulin dose of 0.25 IU/kg was used because GSD-Ia mice have an increased insulin sensitivity[32]. Following an intraperitoneal insulin injection, blood glucose levels in the 53-week-old control mice failed to decrease (Fig. 8b), reflecting an age-related decrease in insulin sensitivity[32,33]. Conversely, following an intraperitoneal insulin injection, blood glucose levels in the 53-week-old 3W-

301L-dosed huR83C mice decreased with time (Fig. 8b), demonstrating that the edited mice were protected against an age-related insulin resistance, consistent with previous observations in the rAAV-treated $G6pc$-/- mice[32].

Histological analysis showed that none of the 3W-301L-dosed huR83C mice developed hepatic tumors at age 53 weeks, similar to the NB-301H-53W mice. Serum levels of aspartate aminotransferase (AST) and alanine aminotransferase (ALT) are biomarkers of liver diseases[34], and both markers have been reported to be elevated in human GSD-Ia patients[35]. The normal ranges for mouse serum AST are reported to be 90.1 ± 8.1 to 293.4 ± 62.7 U/L, and for mouse serum ALT are reposted to be 46.2 ± 5.6 to 239.5 ± 141.2 U/L[36]. At 53 weeks of age, all nine mice edited at age 3 weeks with a low dose of BEAM-301 (3W-301L-53W) were in the normal range for both markers, statistically similar to their age-matched control mice (Fig. 8c). For the mice edited as newborns with a high dose of BEAM-301 (NB-301H-53W), 17 out of 19 had both markers in the normal range, statistically similar to their wild-type control mice (Fig. 8c). In the other 2 mice, expressing 27 and 41 units of hepatic G6Pase-α activity, both AST (360 and 370 U/L) and ALT (323 and 139 U/L) were elevated. The reasons for this were not clear. Neither mouse displayed obvious metabolic abnormalities, although it has been reported that handling a mouse primarily by the body can elevate serum ALT[37].

Enzyme histochemical analysis showed that enzymatically active G6Pase-α in 53-week-old control mice was distributed throughout the liver with significantly higher levels in proximity to blood vessels (Fig. 8d), similar to that observed at 3 weeks of age. In the 53-week-old 3W-301L-dosed huR83C mice, G6Pase-α was also distributed throughout the liver, although less uniformly, with foci containing markedly higher levels of enzymatic activity and other regions harboring little or no G6Pase-α activity (Fig. 8d).

H&E staining showed that both the 53-week-old control and 3W-301L-dosed huR83C mice exhibited no hepatic histological abnormalities (Fig. 8e). Oil red O staining showed that the 53-week-old control mice displayed a variable degree of neutral fat storage (Fig. 8e). Among the six control mice analyzed, four displayed high levels of neutral fat storage. In contrast, among the six 3W-301L-dosed huR83C mice, only the edited mouse expressing 2.6 units of hepatic G6Pase-α activity showed high levels of neutral fat storage (Fig. 8e), while the other five edited mice showed little or no hepatic fat storage. This is consistent with the observation that hepatic triglyceride levels in the 3W-301L-53W mice were significantly lower than their littermate controls (Fig. 7g). In summary, at age 53 weeks, all surviving BEAM-301-dosed huR83C mice displayed a near normal metabolic phenotype, lacked hepatic tumors, and were leaner and protected against age-related insulin resistance.

## Discussion

GSD-Ia is an autosomal recessive, monogenic disease caused by pathogenic variants in the $G6PC1$ gene. The encoded protein, G6Pase-α, is embedded within the endoplasmic reticulum membrane of primarily liver and kidney cortex[1-3]. Previous rAAV-mediated $G6PC1$ gene augmentation studies have shown efficacy in preclinical models of GSD-Ia[9-12] and have been translated into an ongoing phase III clinical trial (NCT05139316). However, there is insufficient data to determine the long-term durability of an episomal-based gene therapy in the liver[13,14]. Gene editing may offer a more durable therapy. To date, both a CRISPR/Cas9-based HDR strategy[17] and a CRISPR/Cas9-based NHEJ strategy[18] have been evaluated in a mouse model of GSD-Ia carrying a prevalent GSD-Ia variant, G6PC1-p.R83C. While both strategies were capable of relieving symptoms of the disease, the editing efficiencies of both approaches were less than 2%[17,18]. In the present study, base-editing was explored as a precise targeting strategy[24-29] to correct the G6PC1-p.R83C variant. Previous studies have used a mouse model in which the G6PC1-p.R83C variant was inserted into the coding sequence

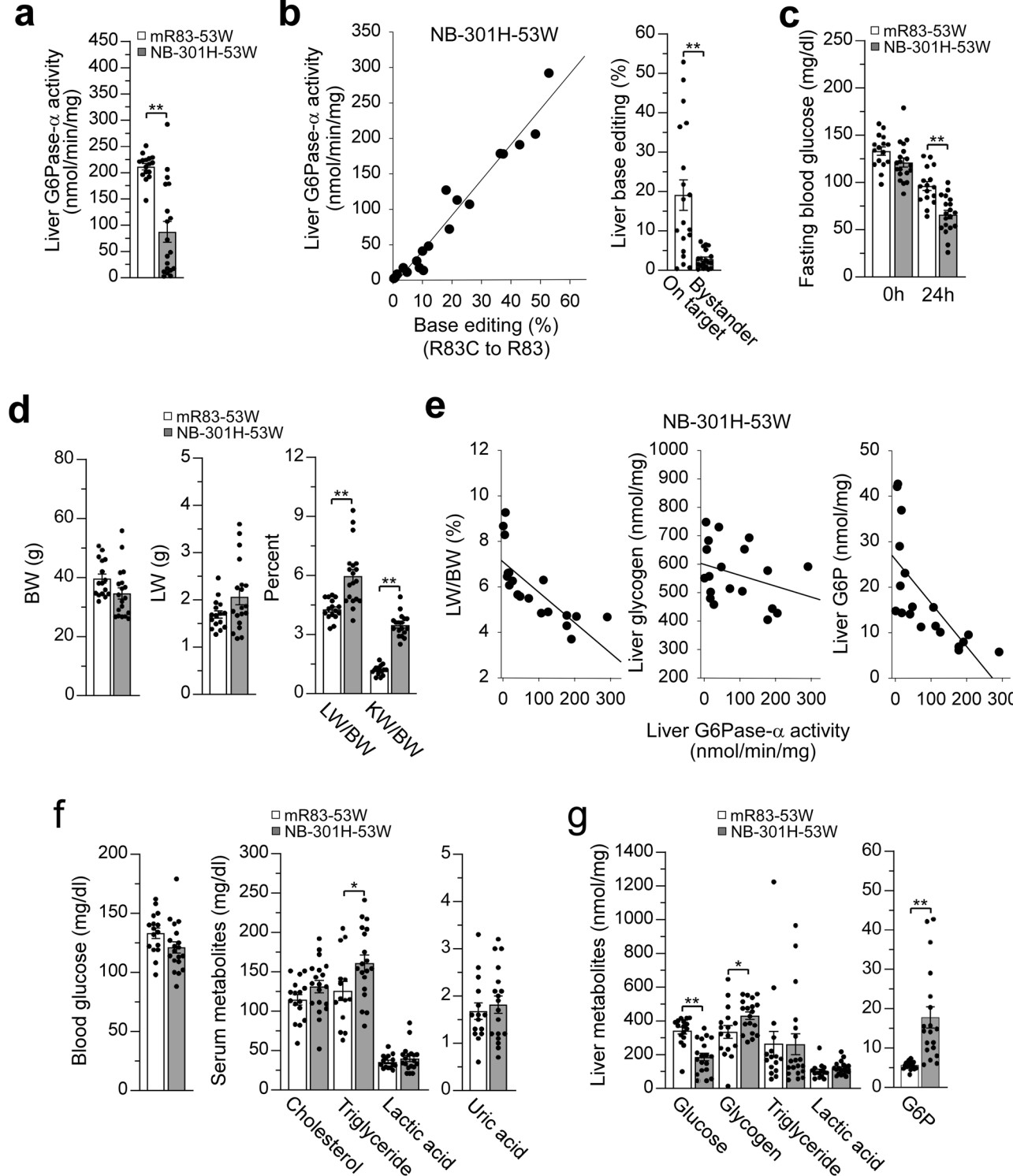

**Fig. 6 | Phenotypic analysis of NB-301H-dosed huR83C mice at age 53 weeks.** Newborn (NB) huR83C mice, non-fasted, were treated with 301H (BEAM-301 at 1.5 mg/kg) and the phenotype of the NB-301H-dosed mice (NB-301H-53W, $n = 19$) was evaluated at 53 weeks of age using sex-matched wild-type littermates (mR83-53W, $n = 16$) as the controls. **a** Liver microsomal G6Pase-α activity. **b** Restoration of hepatic G6Pase-α activity as a function of base editing efficiency along with on-target and bystander values of liver base editing. **c** Fasting blood glucose levels. **d** BW, LW/BW, and KW/BW values. **e** Restoration of hepatic G6Pase-α activity as a function of LW/BW values and hepatic levels of glycogen and G6P in the edited mice. **f** Blood glucose and serum cholesterol, triglyceride, lactate, and uric acid levels. **g** Liver glucose, glycogen, triglyceride, lactate, and G6P levels. Statistics were performed using a two-tailed unpaired $T$ test. Data are presented as Mean values ± SEM, and individual data points for each animal are displayed. * denotes $p < 0.05$, ** denotes $p$ value $< 0.005$.

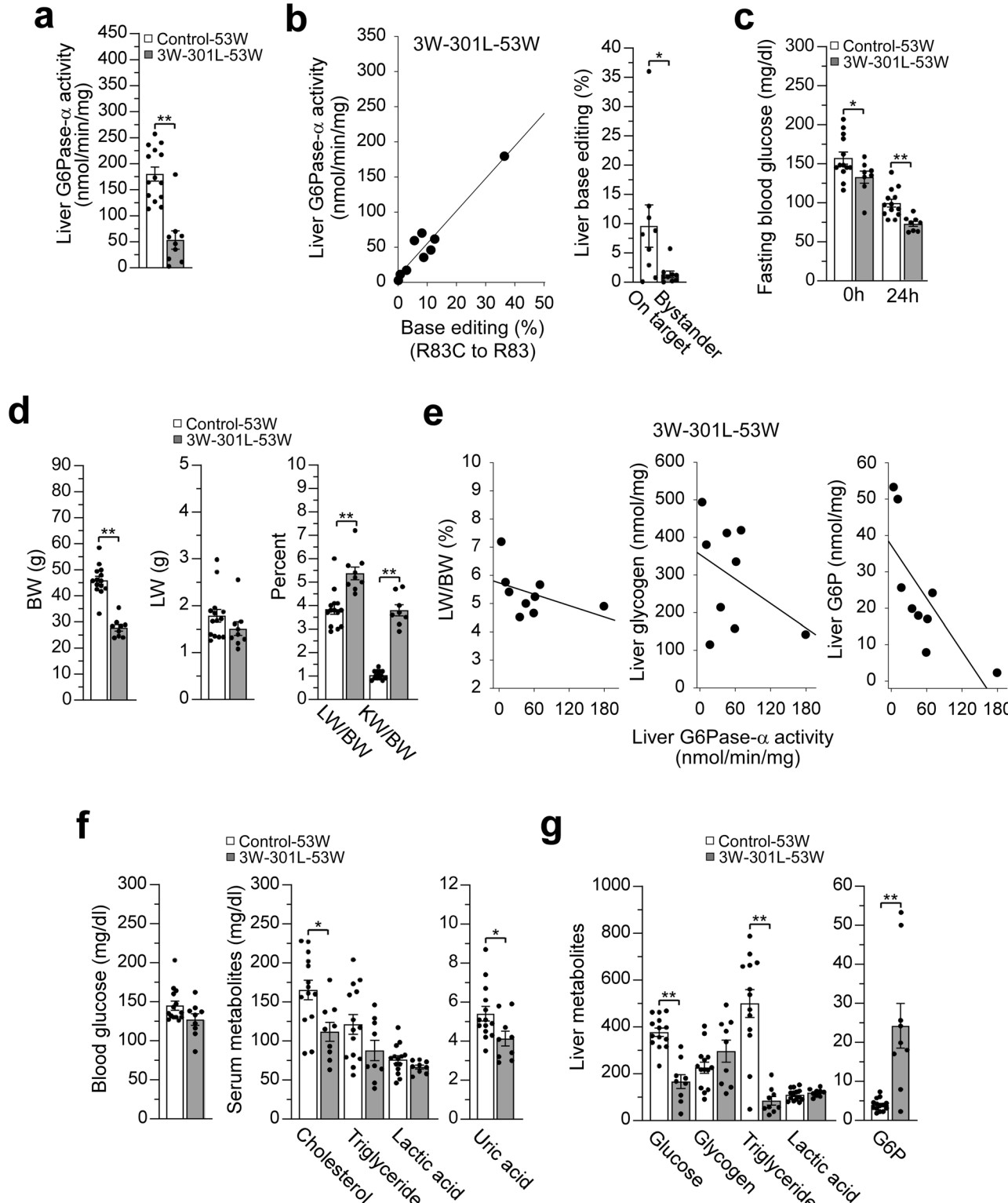

**Fig. 7 | Phenotypic analysis of 3W-301L-dosed huR83C mice at age 53 weeks.**
Three-week-old (3W) huR83C mice, non-fasted, were treated with 301L (BEAM-301 at 0.75 mg/kg) and the phenotype of the 3W-301L-dosed mice (3W-301L-53W) was evaluated at 53 weeks of age using the sex-matched mR83-53W and mR83/huR83C-53W littermates (Control-53W) as the controls. **a** Liver microsomal G6Pase-α activity in Control ($n = 14$) and 3W-301L-53W ($n = 9$) mice. **b** Restoration of hepatic G6Pase-α activity as a function of base editing efficiency along with on-target and bystander values of liver base editing ($n = 9$). **c** Fasting blood glucose levels in Control-53W ($n = 13$) and 3W-301L-53W ($n = 8$) mice. **d** BW, LW/BW, and KW/BW values in Control ($n = 14$) and 3W-301L-53W ($n = 9$) mice. **e** Restoration of hepatic G6Pase-α activity as a function of LW/BW values and hepatic levels of glycogen and G6P in the edited mice ($n = 9$). **f** Blood glucose and serum cholesterol, triglyceride, lactate, and uric acid levels in Control ($n = 14$) and 3W-301L-53W ($n = 9$) mice. **g** Liver glucose, glycogen, triglyceride, lactate, and G6P levels (nmol/mg) in Control ($n = 14$) and 3W-301L-53W ($n = 9$) mice. Statistics were performed using a two-tailed unpaired $T$ test. Data are presented as Mean values ± SEM, and individual data points for each animal are displayed. * denotes $p < 0.05$, ** denotes $p$ value $< 0.005$.

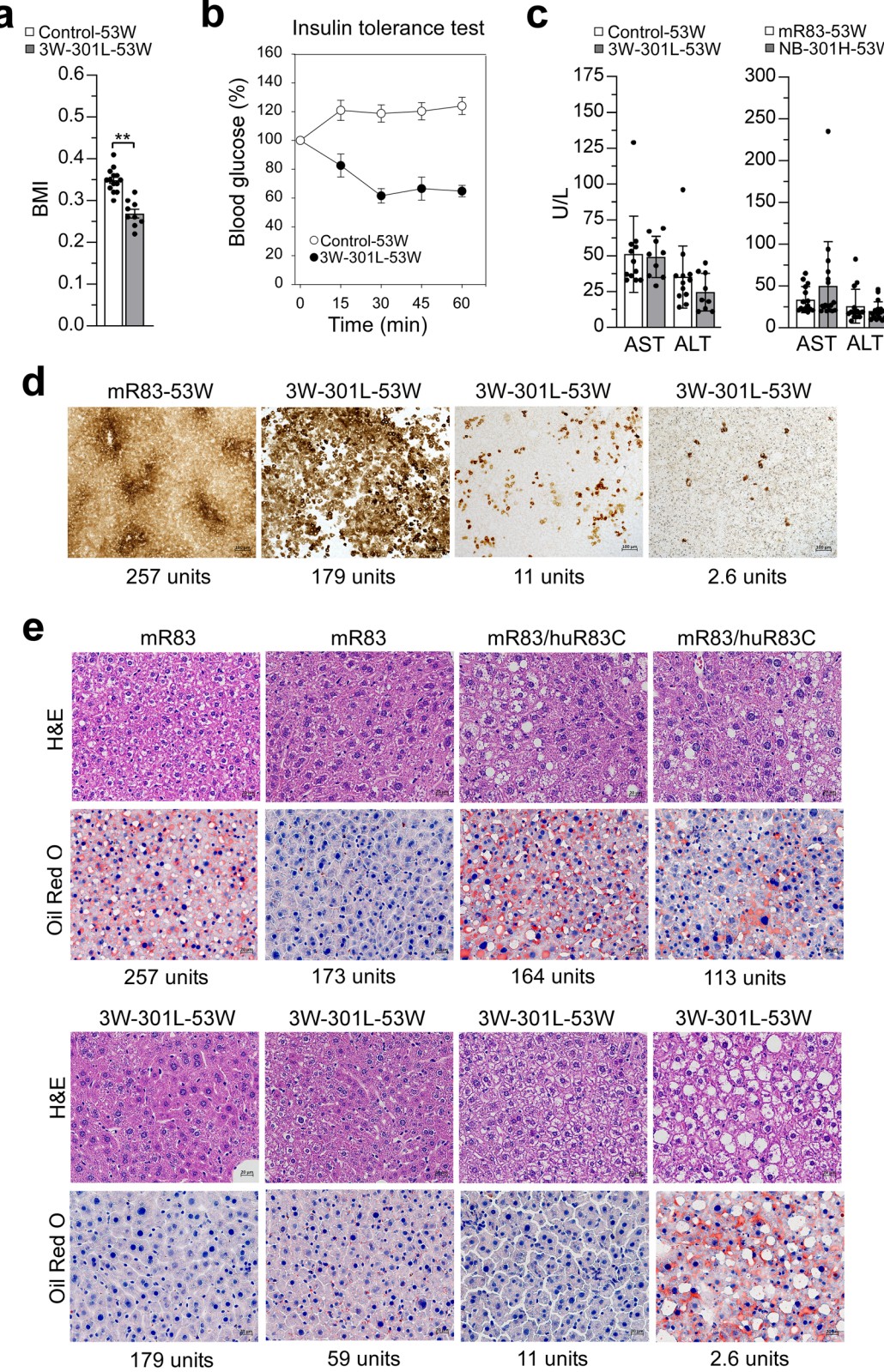

of the mouse *G6pc* gene. In this study, a humanized mouse model, huR83C, was created that contains the entire human *G6PC1-c.247C > T/* p.R83C (*G6PC1*-R83C) coding sequence inserted into the native mouse locus, disrupting the endogenous mouse *G6pc* gene expression. We show that the huR83C mice lack hepatic and renal G6Pase-α activity and manifest a phenotype of impaired glucose homeostasis mimicking GSD-Ia patients[1–3], characterized by fasting hypoglycemia, growth

retardation, hepatomegaly, nephromegaly, hypertriglyceridemia, hypercholesterolemia, and hyperuricemia. As anticipated from studies of the global *G6pc-/-* mice[23], the huR83C mouse had a 39% survival rate at 3-weeks of age, making it a valuable new model for preclinical translation studies of GSD-Ia.

Using the huR83C mice, we have evaluated the efficacy of a high and low dose (301H and 301L) of BEAM-301, an LNP formulation of

**Fig. 8 | Biochemical analysis of 3W-301L-dosed huR83C mice at age 53 weeks.** Three-week-old (3W) huR83C mice, non-fasted, were treated with 301L (BEAM-301 at 0.75 mg/kg) and the biochemical phenotype of the 3W-301L-dosed mice (3W-301L-53W), was evaluated at 53 weeks of age using the sex-matched mR83-53W and mR83/huR83C-53W littermates (Control-53W) as the controls. **a** BMI values in Control-53W ($n = 14$) and 3W-301L-53W ($n = 9$) mice. **b** Insulin tolerance test in Control-53W ($n = 11$) and 3W-301L-53W ($n = 8$) mice. A reduced insulin dose of 0.25 IU/kg was used because GSD-Ia mice have an increased insulin sensitivity[32]. **c** ALT and AST values in Control-53W ($n = 12$), 3W-301L-53W ($n = 9$), mR83-53W ($n = 15$) and NB-301H-53W ($n = 17$) mice. **d** Histochemical analysis of liver G6Pase-α activity in Control-53W ($n = 6$) and 3W-301L-53W ($n = 6$) mice. Each image represents an individual mouse. Scale bar = 100 μm. The numbers represent hepatic G6Pase-α activity expressed in the mice. **e** Hematoxylin and eosin (H&E) and Oil Red O staining of the liver sections in Control-53W ($n = 6$) and 3W-301L-53W ($n = 6$). The liver in the six pairs of experimental animals were examined. A single image from an individual mouse, representative of the results in all mice, is shown to illustrate the results. Scale bar = 20 μm. The numbers represent hepatic G6Pase-α activity expressed in the mice. Statistics were performed using a two-tailed unpaired $T$ test. Data are presented as Mean values ± SEM, and individual data points for each animal are displayed. * denotes $p < 0.05$, ** denotes $p$ value $< 0.005$.

base-editing reagents, to correct the *G6PC1*-R83C variant and associated metabolic abnormalities, through 53 weeks of age. Both NB and 3W huR83C mice were treated with a single, systemic administration of BEAM-301 that resulted in a range of hepatic adenine base editing rates. The edited mice maintained glucose homeostasis without hypoglycemic seizures, manifested an improved metabolic phenotype, and survived to 53 weeks of age. The level of hepatic G6Pase-α activity restored in the huR83C mice correlated to base-editing efficiency. Notably, in individual huR83C mouse, up to ~60% gene allele correction and wild-type hepatic G6Pase-α activity could be restored via BEAM-301. Hepatic G6Pase-α expression was durable and readily exceeded the minimum activity previously shown to be required for metabolic correction and the prevention of HCA/HCC development[9–12]. Compared to mice dosed when newborn, the mice dosed at age 3 weeks displayed higher, less variable levels of hepatic G6Pase-α activity with a liver phenotype more closely approximating that of the control mice, suggesting that administration of BEAM-301, hepatic delivery of mRNA/gRNA, and/or base editing of *G6PC1*-R83C may be impaired in newborn mice. These results are consistent with published reports of LNP-mediated delivery in neonatal mice[38].

We evaluated guide-RNA-dependent off-target editing in primary human hepatocytes and identified only one intronic guide-dependent off-target site, demonstrating the precision of BEAM-301. Based on an integrated risk analysis, an edit at this off-target site is unlikely to impact the function of any associated genes. Therefore, the overall risk of base editing at this off-target site is considered negligible and supports the continued development of BEAM-301 as an investigational treatment for GSD-Ia patients who harbor the p.R83C variant.

As anticipated, systemic infusion of BEAM-301 failed to restore renal G6Pase-α expression, as LNP targeted delivery and editing has been shown to largely occur in hepatocytes within the liver. However, restoration of hepatic G6Pase-α expression in the BEAM-301-edited mice improved nephromegaly. Studies have shown that good metabolic control improves renal function and leads to regression of HCAs in human GSD-Ia patients[39–41]. Our results suggest that by improving metabolic control, BEAM-301 has the potential to alleviate renal dysfunction and regress preexisting HCAs. However, in the absence of gene editing in the kidney, renal impairment could continue to develop over time particularly in aged mice. Rocca et al. have shown that retrograde renal vein injection of a rAAV9 vector efficiently targets the kidney cortex and medulla[42], suggesting that retrograde renal vein administration of BEAM-301 to the huR83C mice may be worth exploring.

These preclinical base editing outcomes compare favorably with the preclinical outcomes of rAAV-*G6PC1*-mediated gene augmentation studies[9–12]. In the current study we show remarkable durability of normal glucose homeostasis in treated huR83C mice out to 53-weeks of age irrespective of dosing BEAM-301 shortly after birth or at 3 weeks of age.

Liver microsomal G6Pase-α background activity in 3-week-old untreated *G6pc-/-* mice varies from 0.5 to 2.2 units. However, the distinction between untreated *G6pc-/-* mice and those rAAV-*G6PC1*-treated *G6pc-/-* mice with a low level of hepatic G6Pase-α activity is the ability to survive long term without glucose therapy. Prior studies have shown that *G6pc-/-* mice cannot survive to 8 weeks of age even under a glucose therapy[23]. In contrast, studies of 25 *G6pc-/-* mice titrated with rAAV-*G6PC1* to express <5 units (0.8 to 4.1 units) of hepatic G6Pase-α activity[11,43,44], all survived the 50-75-week study and maintained glucose homeostasis capable of sustaining 24 hours of fasting. Notably, 10 of the 25 treated *G6pc-/-* mice expressed <2 units of hepatic G6Pase-α activity, the lower limit of quantification of the G6Pase-α assay. Taken together, the 53-week survival of the NB-301H-edited and 3W-301L-edited huR83C mice expressing 2.4 and 2.6 units of hepatic G6Pase-α activity, respectively, and their ability to withstand a fast, was not surprising, despite the near-background level of activity measured.

Interestingly, the 53-week-old edited huR83C mice were leaner and protected against age-related insulin resistance. Kim et al. have previously shown that rAAV-G6PC1-treated *G6pc-/-* mice reconstituted with G6Pase-α activity in the liver but neither the kidney nor the intestine are protected against age-related obesity and insulin resistance[32]. The authors showed that activation of pathways including the hepatic carbohydrate response element binding protein signaling, the NADH shuttle system, and the AMP-activated protein kinase/sirtuin 1/peroxisome proliferator-activated receptor-γ coactivator 1α signaling, contribute to this phenotype[32]. This suggests that similar mechanisms may contribute to the lean and insulin sensitive phenotype of the 53-week-old BEAM-301-edited huR83C mice. Future studies directed towards analyzing the roles of endocrine signals from G6Pase-α-deficient kidney cells and/or enterocytes, that could also contribute to the mice with a lean and insulin sensitive phenotype, could provide additional insight.

Glucose entering the liver is phosphorylated by glucokinase to G6P[45]. In gluconeogenic organs, there are multiple competing pathways utilizing intracellular G6P, including: G6Pase-α-mediated glucose production; glycolysis; the hexose monophosphate shunt; and glycogen synthesis. We have shown that hepatic G6Pase-α deficiency mediates reprogramming of G6P metabolism in GSD-Ia mice[31]. However, the impact of varying levels of hepatic G6Pase-α activity on the various individual G6P utilizing pathways in the liver have not been carefully investigated. The BEAM-301-edited mice displayed variable levels of hepatic G6Pase-α activity which could lead to variable levels of the G6P metabolic pathways, which may explain the lack of correlation between levels of hepatic glucose and hepatic G6P/glycogen accumulation. Future studies titrating the effects of varying levels of hepatic G6Pase-α activity on pathways of G6P utilization would be of value.

In summary, we have generated a murine model of GSD-Ia expressing the human *G6PC1*-R83C transcript and shown that the huR83C mice manifest a phenotype of impaired glucose homeostasis mimicking human GSD-Ia. Using these mice, we showed that BEAM-301-mediated gene editing could correct the pathogenic *G6PC1*-R83C variant sufficiently to normalize their metabolic phenotype and support long-term survival. Our study provides proof of concept for base editing to correct a prevalent pathogenic *G6PC1* variant in a mouse model, offering a variant-specific therapeutic option that results in long-term, efficient, and potentially permanent, correction of the GSD-Ia phenotype.

## Methods

### Generation of the huR83C mouse

The heterozygous mR83/huR83C mice carrying one mouse (m) *G6pc*-R83 and one human (hu) *G6PC1*-R83C allele were created using CRISPR/Cas9-mediated gene insertion technology that disrupted the endogenous mouse *G6pc* gene in the C57BL/6 background (Supplementary Fig. 1). A human cDNA encoding the open reading frame for *G6PC1-c.247C > T* (*G6PC1*-R83C) were inserted into exon 1 of the mouse *G6pc* gene at the ATG start codon (Taconic Biosciences) in a way that created a premature STOP codon in the coding sequence of the mouse *G6pc* exon 1. The insertion placed the human *G6PC1*-R83C cDNA under the control of the mouse *G6pc* promoter/enhancer. Sequence analysis showed that the newly introduced *G6PC1*-R83C allele carried an additional p.V332L mutation. Transient expression assays in COS1 cells[30] showed that phosphohydrolase activity of the G6Pase-α-V332L variant retained 81% of wild-type G6Pase-α activity (Supplementary Fig. 3). The difference in phosphohydrolase activity between huG6Pase-α-wild-type and huG6Pase-α-V332L was not statistically significant.

The mR83/huR83C mice in the C57BL/6 background were used to initiate the breeding colony. We have previously found that the global *G6pc*−/− mice in either the C57BL/6 or the 129S4/SvJaeJ background have a very low survival rate, but survival improves significantly in the mixed background of C57BL/6 and 129S4/SvJaeJ. We therefore crossed to create mR83/mR83 (wild-type), mR83/huR83C (heterozygote), and huR83C/huR83C (homozygote) mice in the mixed C57BL/6/129S4/SvJaeJ (C57BL/129) background which were designated as mR83, mR83/huR83C, and huR83C mice, respectively.

To ensure that the huR83C mice used in this study were in a mixed background consisting of approximately 50% C57BL/6 and 50% 129S4/SvJaeJ (C57BL-50/129-50), we applied a stringent mating strategy. We first mated heterozygous (huR83C/mR83) mice in 100% C57BL background with wild-type (mR83/mR83) mice in 100% 129S4/SvJaeJ background, yielding heterozygous and wild-type mice in a C57BL-50/129-50 background. The huR83C mice in the C57BL-50/129-50 background were then obtained by mating heterozygous mice in the C57BL-50/129-50 background. Progenies from the same litter or the same breeding cage were never mated together.

### Synthesis of mRNA

The mRNA encoding the ABE (Supplementary Data 1) was produced by in vitro transcription from a template plasmid, as previously described[26]. Template plasmids encoded a T7 promoter, a 5′ UTR, a base editor open reading frame, a 3′ UTR, and a 120 polyA tail followed by a type II restriction enzyme site. Plasmids were prepared by ZymoPURE II plasmid kits (D4200; Zymo Research Corporation), linearized by restriction digestion, and purified by phenol-chloroform extraction. Transcription reactions were performed using the NEB HiScribe T7 High-Yield RNA synthesis kit (E2040S; New England Biolabs) with CleanCap AG reagent (N-7113; Trilink Biotechnologies) and included complete substitution of N1-methylpseudouridine triphosphate for uridine-5′-O-triphosphate (UTP). Transcripts were purified by lithium chloride precipitation.

### Cell culture and transfection

Primary human hepatocytes (F00995-P; BioIVT) were seeded in CP medium (Z99029; BioIVT) onto a collagen type I coated 24-well plate (356408; Corning) at a density of 350,000 cells/well and incubated at 37 °C, 5% $CO_2$ for several hours to generate an adherent cell monolayer. Primary hepatic co-cultures were generated by supplementing the cell monolayer with fibroblast cells (EF3003, Kerafast), according to the methods in Bale et al.[46] Given lack of patient-derived hepatocytes, lentiviral transduction was performed to introduce the target site using ultra-purified recombinant lentivirus (Lot 220807LVB01; Vector Builder, Inc) packaging a copy of *G6PC1* that encodes the *c.247C > T* mutation, at an MOI 40. MessengerMAX transfection reagent (LMRNA015; ThermoFisher Scientific) co-formulated with base editor mRNA and gRNA (sequences are provided in Supplementary Data 1) at a 3:1 ratio was used to transfect cell cultures, according to manufacture specifications. Cells were lysed and harvested for genomic DNA at 7 days post-transfection.

### Lipid nanoparticle formulations

The ABE mRNA and gRNA that target *G6PC1*-R83C (Supplementary Data 1) were co-encapsulated at a 1:1 weight ratio in LNP as previously described[29]. The LNPs were generated by rapidly mixing an aqueous solution of the RNA at an acidic pH with an ethanol solution containing four lipid components: a proprietary ionizable lipid, distearoylphosphatidylcholine, cholesterol, and a lipid-anchored polyethylene glycol (all lipids were obtained from Avanti Polar Lipids and are proprietary). The two solutions were mixed using the benchtop microfluidics device from Precision Nanosystems Inc. Post-mixing, the formulations were dialyzed overnight at 4 °C against 1× Tris-buffered saline, pH 7.4 (20228; Teknova). They were subsequently concentrated down using 100,000 Da molecular weight cutoff (MWCO) Amicon Ultra centrifugation tubes (UFC910024; Millipore Sigma) and filtered with 0.2-μm filters (4602; Pall Corporation). The total RNA concentration was determined using Quant-iT Ribogreen (R11490; Thermo Fisher Scientific); the particle size was determined using the Malvern Panalytical Zetasizer. All in vivo studies described were conducted with a proprietary ionizable lipid and LNP composition. The formulated research-grade BEAM-301 drug product was discontinued during the preparation of this study. Commercially available alternatives to the proprietary ionizable lipid used in BEAM-301 can be obtained from Avanti Research (e.g. ALC-0315; 627 890900) (https://avantiresearch.com/product-category/cationic-lipids-transfection/ionizable-lipids) or by Thermo Fisher Lipofectamine MessengerMax (LMRNA015) transfection for in vitro studies, as demonstrated in Supplementary Fig. 2. Additionally, the drug substance reagents, mRNA and gRNA (Supplementary Data 1), can be custom purchased via TriLink (https://www.trilinkbiotech.com/) and Axolabs (https://www.axolabs.com/en/services/gene-editing/).

### Next generation sequencing analysis

NGS was used to determine the percentage of *G6PC1* alleles in the liver corrected by base editing and measure any other allelic changes in the *G6PC1* coding sequence. Frozen mouse liver tissues were homogenized in Dulbecco's phosphate-buffered saline using an Omni Prep Multi-Sample Homogenizer (SKU 06-021; Omni International) and genomic DNA was isolated in duplicate from liver homogenate using the MagMAX™ DNA Multi-Sample Ultra 2.0 Kit (A36570; Thermo Fisher Scientific) per manufacturer's protocol. The genomic DNA samples were eluted in MagMAX DNA Multi-Sample Ultra 2.0 Elution Solution (Thermo Fisher Scientific Inc.) and quantified using a Lunatic UV/Vis plate reader (Unchained Laboratory). The concentration data were used to normalize the genomic DNA samples to 20 ng/μl, when possible, using MagMAX DNA Multi-Sample Ultra 2.0 Elution Solution.

### Targeted amplicon sequencing and quantification of base-editing by NGS

Amplicon sequencing was performed as previously described[26]. Briefly, genomic loci were amplified in 25-μl PCR reactions using Q5 2× Hot Start Master Mix (New England Biolabs), 0.5 mM human *G6PC1* gene-specific forward and reverse primers: forward, GGGCATTAAACTCCTTTGGG and reverse, AGTCTCACAGGTTACAGGGA for the genomic site; forward, GGTTCCATCTTCAGAGGAAG and reverse, CAGGGAACTGCTTTATCA for the G6PC1-R83C site introduce via lentiviral transduction, and 2 μl of genomic DNA template. Barcoded amplicons were generated in 25-μl PCR reactions using Q5 2× Hot Start Master Mix, 0.5 mM barcode primers, and 2 μl of the prior PCR reaction. Barcoded amplicons (Illumina, Inc.) were combined and purified

via DNA agarose gel extraction (D4008; Zymo Research Corporation) or by SPRIselect bead cleanup (B23318; Beckman Coulter). Final clean libraries were then quantified using a Qubit 4 Fluorometer (Q33238; Thermo Fisher Scientific) and sequenced on an Illumina MiSeq Instrument. The data analysis was performed as previously described[26]. On-target *G6PC1* alleles are the sum of 10G_12G and 12G edits which refer to the sequences 5′-CCAGTATGG**G**C**G**CTGTCCAAA-3′ and 5′-CCAGTATGGAC**G**CTGTCCAAA-3′, respectively. Additional base-edited *G6PC1* alleles are the sum of 6G, 6G_10G_12G, and 6G_12G edits, which refer to the sequences 5′-CCAGT**G**TGGAC**A**CTGTCCAAA-3′, 5′-CCAGT**G**TGG**G**C**G**CTGTCCAAA-3′, and 5′-CCAGT**G**TGGAC**G**CTGTC-CAAA-3′, respectively.

In summary, all targeted NGS data were analyzed by performing 5 steps: 1) Illumina demultiplexing, 2) read trimming and filtering, 3) alignment of all reads to the expected amplicon sequence, 4) generation of alignment statistics and quantification of base-edited alleles, and 5) calculation of the frequencies of corrected *G6PC1* alleles. Since GSD-Ia is an autosomal recessive disorder, the allele frequency represents the combined frequency of heterozygous and homozygous allelic corrections in individual hepatic cells. Only a single allele correction is required in a cell to restore wild-type function.

## Off-target editing analysis

Primary human hepatocytes were harvested from 3 donor livers (BioIVT) and plated on BioCoat Rat Collagen I, 24-well plate (354408; Corning) according to manufacturer's instructions. BEAM-301 was diluted in INVITROGRO CP media (Z990003; BioIVT) and added to the culture wells at the desired concentration. At the terminal timepoint of 1 week, cells were harvested, and genomic DNA was isolated using the *Quick*-DNA™ MicroPrep Plus Kit (D4074; Zymo Research) following the manufacturer's protocol.

Potential off-target sites were identified using ONE-Seq and Digenome-seq. To construct the ONE-seq library, the Cas-OFFinder software tool[47] was run on the GRCh38/hg38 human reference genome using the protospacer sequence and a relaxed PAM requirement (NNGRRN; where "R" stands for "A or G" and N stands for "A, C, G, or T") to account for the PAM preferences of known off-target sites[48]. Sites matching the input pattern with either: (a) 7 or fewer mismatches in the targeted DNA and no bulges or (b) a DNA or RNA bulge (2 bases) and up to 3 mismatches were enumerated. A SurePrint Oligonucleotide Library (G7239A; Agilent) was constructed with 14,509 sites based on mismatch parameters and genomic annotation. The ONE-seq assay was performed on this library as described[49] using a saturating concentration of ribonucleoprotein.

Digenome-seq was performed as described[50] using human genomic DNA isolated from GM11468, a human B-lymphocyte cell line heterozygous for the *G6PC1 c.247C > T* allele (Coriell Institute, Camden, NJ), and a saturating concentration of ribonucleoprotein.

To evaluate the presence of candidate off-target edits in primary human hepatic cells, rhAmpSeq panels were constructed containing: sites nominated from ONE-seq and Digenome-seq; sites identified in silico as low-mismatch sites (but not nominated by either assay); and others (including the 1 on-target site and 20 negative control sites). Genomic DNA was isolated from the primary human hepatocytes treated with BEAM-301 at a concentration at or above the EC$_{90}$ of on-target editing and multiplex amplicon sequencing via rhAmpSeq (Integrated DNA Technologies) was performed as described[51]. The EC$_{90}$ was measured in primary human hepatocytes (PHH) that harbor the target site introduced via lentiviral transduction to enable detection of the on-target edit, since healthy human hepatocytes do not harbor the target allele. Multiplex amplicon sequencing via rhAmpSeq (Integrated DNA Technologies) was performed as described[51]. The rhAmpSeq was also performed on untreated, donor-matched primary human hepatocytes as controls. Sequencing reads were trimmed, stitched, aligned to the human genome reference sequence (GRCh38/

hg38), and filtered for minimum base-quality score of 30 and mapping quality of 10. An odds ratio quantifying the enrichment of each observed variant in the treated sample relative to the untreated sample was calculated, and a Fisher's exact test was used to assess statistical significance based on an empirically determined threshold (Supplementary Data 2).

## Animal studies

Animal studies were conducted either under an animal protocol approved by the Animal Care and Use Committee at *Eunice Kennedy Shriver* National Institute of Child Health and Human Development or an animal protocol approved by the Institutional Animal Care and Use Committee at University of Massachusetts Medical School and CRADL® (Charles River Accelerator and Development Lab) in Cambridge, Massachusetts. All animal procedures have been done according to the institutional guidelines and approved by the local ethics committee. The animal experiments abide by the ARRIVE guidelines (https://arriveguidelines.org/arrive-guidelines). Mice were maintained on a standard NIH-31 Open formula mouse/rat sterilizable diet (Envigo) or Prolab IsoPro RMH 3000 diet (labdiet) without any restrictions. Mice were housed at the animal facility in a 12-hour light/dark cycle, with an ambient temperature controlled within 65–75 °F (18–24 °C) and humidity level between 40–60%. To reduce feeding competition between littermate pups, the size of the litter was usually reduced to 6-8 pups.

Animal welfare monitoring and euthanasia followed the guidance of the NIH ACUC (https://oacu.oir.nih.gov/system/files/media/file/2022-04/b13_endpoints_guidelines.pdf) who approved our endpoints. In brief the key endpoints were: significantly hunched posture, significantly reduced activity or impaired mobility that interferes with normal eating and drinking, significantly rough fur, a weight loss of > 20%, a body condition score of 1 (out of 5), weak or no response to external stimuli, respiratory distress or a pain score of 3 or 4 (out of 4) when therapy is ineffective or needs to be withheld for research reasons. The mice were euthanized by carbon dioxide following the AVMA guidelines for the Euthanasia of Small Laboratory and Wild-Caught Rodents, 2020 Edition (https://www.avma.org/sites/default/files/2020-02/Guidelines-on-Euthanasia-2020.pdf).

BEAM-301 was infused into NB huR83C pups via the temporal vein and into 3W huR83C mice via the retro-orbital sinus. The sex-matched mR83 and mR83/huR83C littermates which have indistinguishable phenotypes were used as controls. Glucose therapy was not provided to BEAM-301-dosed huR83C mice, and the edited mice were not subjected to fasting before euthanasia and tissue collection. Since the untreated huR83C, BEAM-301-dosed huR83C, and control mice used in the study were in the fed state and had access to food ad libitum, their serum glucose levels could vary depending upon the time of study relative to their last food intake.

For the fasting glucose test, the BEAM-301-treated huR83C and control mice were transferred into clean cages without any food but with free access to water for 24 hours. Blood glucose analysis was performed on blood obtained from the tip of the tail at 0 and 24 hours after food deprivation using the HemoCue® Glucose 201 System (HemoCue America) according to manufacturer's instructions.

The BMI value in mice was calculated by dividing the body weight measured in grams by the square of the body length measured in cm. The body length of each mouse was measured as the distance from the tip of the nose to the base of the tail.

For insulin tolerance testing, the 301L-treated huR83C and control mice were subjected to fasting for 4 hours, followed by body weight measurement. Each mouse then received an intraperitoneal injection of 0.25 IU/kg of sterile insulin (#I1882; Sigma-Aldrich) diluted in PBS. Blood glucose was measured at 0, 15, 30, 45 and 60 min after insulin injection using the HemoCue® Glucose 201 System.

## Phosphohydrolase assays

Liver microsome isolation and microsomal phosphohydrolase assays were performed as described previously[23,52]. Permeabilized microsomes were prepared by incubating intact microsomes in sucrose buffer containing 0.2% sodium deoxycholate for 20 min on ice. In phosphohydrolase assays, reaction mixtures (50 µl) containing 50 mM sodium cacodylate buffer, pH 6.5, 2 mM EDTA, 10 mM G6P and 100–200 µg of permeabilized microsomes were incubated at 30 °C for 10 min. The background of non-specific phosphatase activity was estimated by pre-incubating permeabilized microsomes in 20 mM sodium acetate buffer, pH 5 for 10 min at 37 °C to inactivate the acid-labile G6Pase-α. One unit of G6Pase-α activity represents one nmol G6P hydrolysis per minute per mg microsomal protein. The background of non-specific phosphatase activity varied in the lysates from 0.2 to 2 units. The background was subtracted from all data points.

Enzyme histochemical analysis of G6Pase-α was performed on cryopreserved optimal cutting temperature compound (OCT) (#4583; Sakura Finetek USA) embedded tissue sections as described previously[53]. Briefly, 10 µm thick tissue sections were incubated at room temperature for 10 min in a reaction buffer containing 40 mM Tris-maleate pH 6.5, 10 mM G6P, 300 mM sucrose, and 3.6 mM lead nitrate, followed by 2 washes in 300 mM sucrose solution. Then, the tissue sections were incubated for 2 min at room temperature in 0.09% ammonium sulfide solution and the trapped lead phosphate was visualized following conversion to the brown colored lead sulfide. Images of the tissue sections were taken by the Imager A2m microscope with Axiocam 506 camera and the ZEN 2.6 software (Carl Zeiss).

## Measurement of serum metabolites

Blood was collected by cardiac puncture of the non-fasted, anesthetized mice at sacrifice. Blood was transferred into MiniCollect® TUBE 0.8 ml CAT Serum Separator (#450472, Greiner Bio-one), allowed to coagulate for 20 min, then centrifuged for 10 min at 2000 × g. Serum was transferred to a new microtube, flash-frozen in liquid nitrogen, and stored at −80 °C.

Serum cholesterol and uric acid were analyzed using the corresponding Liquid Stable Reagents (# TR13421, #TR24321, ThermoFisher Scientific) or Cholesterol/ Cholesteryl Ester Assay Kit (#ab65359, ABCAM) and Uric Acid Assay Kit (#ab65344, ABCAM). Serum L-Lactate was measured using the EnzyFluo™ L-Lactate Assay Kit (#EFLLC-100, BioAssay Systems), the Lactate Assay Kit (#K627-100, BioVision) or the L-Lactate Assay Kit (#ab65331, ABCAM). Serum triglyceride was analyzed using the Serum Triglyceride Determination Kit (#TR0100, Sigma), the Triglyceride Assay Kit (#K622-100, BioVision) or the Triglyceride Assay Kit (#ab65336, ABCAM).

Serum AST and ALT activity were measured using the EnzyChrom™ Aspartate Transaminase Assay Kit (#EASTR-100, BioAssay Systems) and the EnzyChrom™ Alanine Transaminase Assay Kit (#EALT-100, BioAssay Systems), respectively.

## Measurement of hepatic metabolites

Liver tissues obtained from non-fasted mice were homogenized in 5% NP-40 solution at 10:1 ratio (10 µl 5% NP-40 solution per 1 mg of tissue), deproteinized by incubation at 99 °C for 5 min, and then centrifuged 17,000 × g for 15 min to remove insoluble materials. Glycogen (#K646-100, BioVision), G6P (#K657-100, BioVision), glucose (#K686-100, BioVision; #ab102517, ABCAM) and lactate (#K607-100, BioVision; #ab65331, ABCAM) in the deproteinized liver lysates were measured using the corresponding kits. Calculations of glycogen in nmol/mg were based on a glycogen molar mass of 666.5777 g/mol.

Hepatic triglycerides were isolated by homogenizing liver tissues in 5% NP-40 solution at 10:1 ratio (10 µl 5% NP-40 solution per 1 mg of tissue), heated to and incubated at 85 °C for 5 min and cooled down to room temperature. After repeating once the heating/cooling step, tissue samples were centrifuged for 15 min at 17,000 × g and triglycerides in the supernatant were measured using the Triglyceride Assay Kit (#K622-100, BioVision) or the Triglyceride Assay Kit – Quantification (#ab65336, ABCAM).

## Histopathology

H&E staining was performed on tissues fixed in 10% neutral buffered formalin (#08379, Polysciences) and Oil Red O staining was performed on cryopreserved OCT (#4583; Sakura Finetek USA) embedded tissues processed according to standard procedures. The tissue sections were visualized using the Imager A2m microscope with Axiocam 506 camera and the ZEN 2.6 software (Carl Zeiss).

## Statistics and reproducibility

The statistical analyses reported are two-tailed unpaired T tests performed using the GraphPad Prism Program, version 10.2.2 (GraphPad Software). Data are presented as Mean values ± SEM, and individual data points for each animal are displayed. * denotes $p < 0.05$, ** denotes $p$ value $< 0.005$.

This animal study was performed once. Each animal in the experiment was infused with the editing reagents independently and represents an independent outcome within the experiment. All experiments used the huR83C mouse homozygous for the pathogenic variant *G6PC1*-R83C. Since GSD-Ia is an autosomal recessive disease, every experiment included the mR83/mR83 (wild-type) and mR83/huR83C (heterozygote) mice as the controls. In the shorter 3-week study, untreated huR83C mice, which had a 3-weeks survival rate of 39%, were also included as controls.

## Reporting summary

Further information on research design is available in the Nature Portfolio Reporting Summary linked to this article.

# Data availability

All data supporting the findings described in this manuscript including the mRNA encoding the ABE are available in the Article, Supplementary Information, and Source data file published alongside the paper. The raw sequence reads are available on NCBI under the accession number PRJNA1164309. Source data are provided with this paper.

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

## Acknowledgements

We thank Regina Raz for schematic and description of Supplementary Fig. 1, Raymond Yang and Sneha Gampa for LNP formulation, Faith Musenge, Monique Otero, Edmond Wong, and Zach Howard for in vivo dosing protocol optimization, Kenton Hetrick for description of off-target analysis. Martina Schinke, Rachel Goldsmith, Rodrigo Laureano, and Sunita Goyal for thoughtful review of the manuscript. This research was supported by the Intramural Research Program of the *Eunice Kennedy Shriver* National Institute of Child Health and Human Development, National Institutes of Health, United States Project Code Z HD 00912 2017 searchable at https://intramural.nih.gov/search/index.taf (J.Y.C.). This research was also supported by a Cooperative Research and Development Agreement between BEAM Therapeutics, Cambridge, MA 02142, USA and the *Eunice Kennedy Shriver* National Institute of Child Health and Human Development, National Institutes of Health, United States (J.Y.C.).

## Author contributions

I.A. designed and executed the research, drafted, and edited the manuscript; Y.A.-S. conceptualized study designs, coordinated the study, analyzed the data, and drafted relevant portions of the manuscript; L.Z. and C.L. characterized the G6PC variants, analyzed the data, and edited the manuscript; H.-D.C. and S.G. performed animal studies and analyzed the data. M.S.P. conceptualized study designs, designed/optimized the base-editor, and drafted relevant portions of the manuscript; F.M.G. and G.C. conceptualized the study design; D.L. coordinated the animal study and NGS analysis; S.B. and B.B. performed animal studies; T.P.F. performed biochemical, and NGS analysis; V.H. performed biochemical analysis; L.-I.C. and G.L. optimized the base editor; J.D. performed NGS analysis; J.J.Q. designed and executed the off-target assay; L.Y. designed and analyzed the off-target editing; L.B. conceptually designed the off-target study; L.S.S. designed the ONE-Seq panel, analyzed the data, and performed In-silico analysis; C.B. nominated the Digenome-seq and validated rhAmpSeq assessments; P.K. and P.G. analyzed the Rhamp-seq data or Digenome-seq data; K.W.M. experimentally assessed the One-seq; T.L. performed the off-target studies in primary hepatocytes; B.C.M. analyzed the data and edited the manuscript. J.Y.C. designed the research, acquired the funding, analyzed the data, and wrote the manuscript.

## Funding

## Competing interests

Y.A.-S., M.S.P., G.C., D.L., S.B., B.B., T.P.F., V.H., J.D., L.-I.C., J.J.Q., L.Y., L.B., L.S.S., C.B., P.K., P.G., K.W.M. and T.L. are employees and shareholders of Beam Therapeutics; G.L. and F.M.G. are shareholders of Beam Therapeutics. The remaining authors declare no competing interests.
