## [Transparent Peer Review file · Nature Communications]

Base-editing corrects metabolic abnormalities in a humanized mouse model for glycogen storage disease type-1a

Corresponding Author: Dr Janice Chou

Version 0:

Reviewer comments:

Reviewer #1

(Remarks to the Author)

In this manuscript, the authors utilized CRISPR/Cas9-mediated gene insertion to disrupt the endogenous mouse G6pc gene and insert a human cDNA encoding the open reading frame for G6PC-c.247C>T (G6PC-R83C) into exon 1 of the mouse G6pc gene at the ATG start codon. This resulted in the creation of a humanized mouse model with a premature STOP codon in the mouse G6pc exon 1 and the insertion of human G6PC-c.247C>T (G6PC-R83C). The homozygous mutant mice exhibited a GSD-1a phenotype. Additionally, the authors employed lipid nanoparticles to deliver the messenger RNA of ABE with an saCas9 'NNGRRR' protospacer-adjacent motif preference and a synthetic guide RNA in vivo. The treated mice demonstrated a maximum base-editing efficiency of approximately 60% in the liver and achieved some level of physiological rescue. The manuscript provides pre-clinical study for a potential genome editing therapy for GSD-1a. However, several concerns arise:

1. The G6PC-R83C humanized mouse model has two major flaws. It carries an additional p.V332L mutation. Transient expression of the p.V332L mutation in COS1 cells showed that phosphohydrolase activity of the G6Pase- α -V332L variant retained 81% of wild-type activity. No statistical data was provided to determine if this difference is statistically significant. Could the additional variant be the reason the authors only observed partial physiological rescue even in the high dose group? Another concern is regarding the background of the mice. The authors stated that global G6pc^{-/-} mice have significantly improved survival in the mixed background of C57BL/6 and 129S4/SvJaeJ (C57BL/129) compared to a pure C57BL/6 background and assumed that the same is true for the G6PC-R83C humanized mouse model. Do the authors anticipate that the survival of global G6pc^{-/-} mice and G6PC-R83C humanized mice would be even better in the 129S4/SvJaeJ background, or is the mixed background necessary? All the huR83C homozygous mice used in this manuscript were of mixed background, which could be a confounding factor as the mice may exhibit varying levels of the mixed background, resulting in differences in phenotype severity.
2. It is surprising that no off-target analysis was conducted. The authors did not even mention off-target effects in the entire manuscript. This omission is unacceptable. The authors should perform at least one unbiased whole-genome off-target analysis in a cell line or genomic DNA from the cell line.
3. G6pc flox mice have been reported. Are there any studies investigating the knockout of the G6pc gene specifically in the liver or kidney? Corrective editing was only detected in the liver, not the kidney in this manuscript. Therefore, I am curious to know if there are any pathogenic physiological effects in mice with only kidney deletion of the G6pc gene.
4. The authors observed a wide range of editing efficiencies. NB-high dosed mice exhibited a range in base-editing efficiency of 0.8% to 35%, while 3W-high dosed mice varied with a range in base-editing efficiency of 2.9% to 36.6%. The variability in the NB group could be attributed to technical challenges of tail vein injection, but retro-orbital injection for the 3-week group should be easier to perform. Do the authors have any insights into why there is so much variability?
5. What is the lowest editing threshold required to achieve therapeutic benefit for GSD-1a patients?
6. Did the authors check the AST/ALT levels in the treated mice?
7. In the long-term study, no premature death was observed in the NB-301H-edited huR83C mice. The editing rates ranged from ~0.5% to 53%. The mouse with the lowest hepatic G6Pase- α activity was recorded at 2.4 units in the NB-dosed huR83C mice. As the authors pointed out, the lowest detection level of quantitation for the microsomal G6Pase- α assay is 2 units. Therefore, this particular mouse exhibited G6Pase activity at knockout levels. How did it survive for 53 weeks and overnight fasting? Similar findings were observed in the 3W-L-dosed huR83C mice, where one edited mouse exhibited only 2.6 units of hepatic G6Pase- α activity.

8. The summary provided in lines 399-401 is highly misleading. The authors reported a 53-week survival rate of 69% among the 21 huR83C mice treated with 301L at age 3 weeks, claiming that all BEAM-301-dosed huR83C mice displayed a near-normal metabolic phenotype and were leaner and protected against age-related insulin resistance is inaccurate and misleading.

Reviewer #2

(Remarks to the Author)

In this manuscript, Arnaoutova et al. present a straightforward, systemic, elegant, and experimentally sound study in which base-editing is used to correct glycogen storage disease type 1a in mice. To my knowledge, this work provides the first preclinical proof of concept for this state-of-the-art methodology for the treatment of a monogenetic metabolic disease. Find below a number of points that, when addressed, in my view will further improve the quality of this manuscript.

Strong points in the manuscript:

- Major strengths of the study are that it uses a humanized model of GSD Ia (by insertion of the entire human G6PC1-R83C coding sequence and disabling native mouse G6pc expression), and that it uses a more efficient gene-editing tool (CRISPR-based base editing) as compared to the authors' previous studies (using either homologous recombination or a CRISPR/Cas9-based double-strand oligonucleotide insertion strategy)
- Clear description of the elaborate analysis of the different possibilities and eventual results from the administration of BEAM-301 in Figure 1.
- Clear and extensive description of results in both NB and 3-week old mice, wild-type or heterozygous, with low or high doses of BEAM-301, and analyzed at different time points (3 weeks (for NB only), 8 weeks, and 53 weeks). However, some questions and points for improvement remain (see below).

Questions/points-for-improvement

Major

- The authors mention they monitor phenotypic correction through one year after base-editing-mediated correction of the mutated G6PC. However, liver tumor formation, one of the major long-term complications of hepatic GSD Ia in both humans and mice, usually only develop around the time and after end point of the study presented in this manuscript. Could the authors explain why did not follow-up for a longer period to assess prevention of liver tumor formation? In addition, why did the authors only include NB-301H and 3W-301L treated groups, and not also (the NB-301L and) 3W-301H groups?
- Data hepatic (pre)-tumor incidence, or liver tumor markers at the 53-week endpoint are not included for all 9 mice of the 3W-301L-dosed huR83C group. As HCA/HCC is a common and severe complication observed in correct glycogen storage disease type 1a patients and mouse models, it is valuable to include such data to assess the effect of the treatment on HCA/HCC susceptibility.
- The authors state that their results support the development of BEAM-301 as a potential therapeutic for patients with GSD-Ia carrying the G6PC-R83C variant. Can the authors elaborate shortly if this only applies to patients homozygous for this variant, or also for patient where only one of the mutated alleles contains this variant?
- Line 92-93: "The prevalent pathogenic G6PC-R83C variant contains a single G>A transition mutation." However, earlier in the abstract and in the remainder of the work the authors state the mutation is G6PC-g.247C>T, which suggest it is a C>T mutation. Are we correct to assume the authors in lines 92-93 refer to the opposing strand?
- Lines 124-125: "the mR83 and mR83/huR83C littermates display indistinguishable wild-type phenotypes and, therefore, were used as controls". Do the authors have data to prove that they indeed have indistinguishable phenotypes? We do agree, based on previous studies by the authors using murine GSD Ia models, that wild-type and heterozygous G6pc-deficient mice are similar in terms of key GSD Ia hallmarks such as survival rate, tolerance to fasting etc, but other data indicates that some hepatic parameters, such as (mild) glycogen accumulation, may still be present in mice with reduced but substantial remaining G6PC activities (PMID: 34157136).
- Do the authors have an explanation why BEAM-301 treatment does not result in increased G6Pase activity in the kidney cortex (Fig. 2, 4)? The authors speculate in lines 447-453 in the discussion, but do they have any suggestions for gene editing delivery tools more suitable for the kidney? What about long-term complications in the kidneys of treated mice after 53 weeks?
- There appears to be a discrepancy in blood glucose results in NB-301H-3W mice presented in Figure 2C versus 3A, with glucose levels being significantly lower in these animals versus controls in Figure 3A only. The authors should discuss and explain this difference.
- In Figure 4G, liver glucose levels are lower in NB-301H-8W, but not in 3W-301H-8W mice as compared to controls. Yet, the degree of G6P and glycogen accumulation in the liver is similar in these 2 groups. The authors should discuss these different phenotypes.
- Data presented in Figure 8 show that after 53 weeks, 3W-301L-dosed huR83C mice are leaner and more insulin sensitive compared to control mice. Given that renal (and likely intestinal) G6PC activity remains uncorrected in these animals, it can be hypothesized that that (e.g. endocrine signals from) G6PC deficient kidney cells and/or enterocytes contribute to this lean, insulin-sensitive phenotype. The authors should consider and discuss this possibility.

Minor

- The gene/protein name for human G6PC should be corrected to G6PC1
- In the abstract, the authors state "BEAM-301, lipid nanoparticles 34 containing guide RNA and mRNA encoding a newly-engineered adenine base editor". However, to my knowledge the term 'guide RNA' is often used in combination with gene editing using the CRISPR/Cas9 system, and I presume the authors actually mean a 'repair template' to correct the mutated G6PC. If this is correct, could the authors adjust this?
- Lines 60-62: "In addition, dietary therapies do not address the underlying pathological processes and long-term

complications, so hepatocellular adenoma/carcinoma (HCA/HCC) and renal disease still occur in metabolically compensated patients". This is indeed true, but the authors fail to highlight that dietary compliance and associated good metabolic control can at least reduce the incidence of these long-term complications (eg PMID 25308557, 28568353, 28612263), which may be good to highlight

- Line 69: "many patients have pre-existing antibodies to the rAAV vector"; can the authors strengthen this claim by a reference?

- Fig. 2E: label on y-axis does not seem correct (relative values is indeed true for the organ/body weight ratios, but not for the body weight data (which is in grams, an absolute unit))

- Fig. 3B: Hepatic glycogen accumulation is quantified as nmol/mg; however, the molar mass of glycogen depends on the extent of its branched structure. Can the authors explain what value was used for this / how the quantification was performed?

- Fig. 3C: H&E staining on liver of NB-301H-3W with correction to 33 units still shows a bit of vacuolopathy (enlarged hepatocytes) that usually indicates glycogen and/or fat accumulation, yet in Fig. 3C the authors showed that hepatic levels of these metabolites in 301H mice are completely restored to wild-type levels. Can the authors explain the yet remaining slight enlargement of the hepatocytes?

- The authors note in Fig. 4 a reduced editing variability and increased editing efficiency when performed at 3W rather than in NB mice. Do the authors have potential explanations for this? And how would this impact clinical translation? Eg correcting the genetic defect earlier in life would potentially benefit the patient more (especially right after birth when demand for gluconeogenesis is high) vs potential reduced editing efficiency

- Please indicate more clearly in the legends the fed/fasting state in which the parameters were analyzed (especially relevant for interpreting serum metabolite levels in Figures 3-4); eg. It looks like data in Fig. 4F is from fed mice, but in order to interpret GSD Ia biochemical symptoms (eg fasting hypoglycemia, fasting hypertriglyceridemia etc) it is also important to analyze these parameters in a fasted state. When carefully reading the methods section this becomes evident, but as the nutritional state is critical in this disease context, we would like to advise to also state this more clearly in the figure legends.

- Fig. 5: The authors note that of the 21 3-week old 301L mice, 18 survived, yet G6Pase activity was only assessed in 8 of them; is there any specific reason?

- Do the authors have a potential explanation for the marked lower hepatic TG levels in 3W-301L-53W mice compared to controls (Fig. 7G, 8D)?

- Fig. 8A: How is BMI calculated in mice? (especially the length parameter)

Reviewer #3

(Remarks to the Author)

Version 1:

Reviewer comments:

Reviewer #1

(Remarks to the Author)

The authors have addressed all my comments, and I have no further concerns.

Reviewer #2

(Remarks to the Author)

The authors have satisfactorily addressed our comments and suggestions, with one minor issue remaining. The authors have, as requested, now adapted the human gene/protein name for G6PC1 from G6PC to G6PC1. However, to our knowledge the official gene name for mice is still G6pc (without 1). Could the authors adapt and make sure G6PC1 is used when referring to human, and G6pc when referring to mouse?

Reviewer #3

(Remarks to the Author)

Base-editing corrects metabolic abnormalities in a humanized mouse model for glycogen storage disease type-Ia (NCOMMS-24-19638)

Reviewer #1 (Remarks to the Author):

In this manuscript, the authors utilized CRISPR/Cas9-mediated gene insertion to disrupt the endogenous mouse G6pc gene and insert a human cDNA encoding the open reading frame for G6PC-c.247C>T (G6PC-R83C) into exon 1 of the mouse G6pc gene at the ATG start codon. This resulted in the creation of a humanized mouse model with a premature STOP codon in the mouse G6pc exon 1 and the insertion of human G6PC-c.247C>T (G6PC-R83C). The homozygous mutant mice exhibited a GSD-Ia phenotype. Additionally, the authors employed lipid nanoparticles to deliver the messenger RNA of ABE with an saCas9 'NNGRRT' protospacer-adjacent motif preference and a synthetic guide RNA in vivo. The treated mice demonstrated a maximum base-editing efficiency of approximately 60% in the liver and achieved some level of physiological rescue. The manuscript provides pre-clinical study for a potential genome editing therapy for GSD-Ia. However, several concerns arise:

1. The G6PC-R83C humanized mouse model has two major flaws. It carries an additional p.V332L mutation. Transient expression of the p.V332L mutation in COS1 cells showed that phosphohydrolase activity of the G6Pase- α -V332L variant retained 81% of wild-type activity. **No statistical data was provided to determine if this difference is statistically significant. Could the additional variant be the reason the authors only observed partial physiological rescue even in the high dose group?** Another concern is regarding the background of the mice. The authors stated that global G6pc^{-/-} mice have significantly improved survival in the mixed background of C57BL/6 and 129S4/SvJaeJ (C57BL/129) compared to a pure C57BL/6 background and assumed that the same is true for the G6PC-R83C humanized mouse model. **Do the authors anticipate that the survival of global G6pc^{-/-} mice and G6PC-R83C humanized mice would be even better in the 129S4/SvJaeJ background, or is the mixed background necessary?** All the huR83C homozygous mice used in this manuscript were of mixed background, which could be a confounding factor as the mice may exhibit varying levels of the mixed background, resulting in differences in phenotype severity.

Response:

We thank the reviewer for the critical reading and support of the paper.

The p.V332L variant: We were aware of the additional p.V332L variant in the G6PC1-R83C humanized mouse model (huR83C) and examined its effect on the mouse phenotype. We performed two independent transient expression studies, each in triplicate, of the pSVL-huG6Pase- α -V332L construct in COS-1 cells using the pSVL-huG6Pase- α -wild-type construct as the control. Our studies showed that the averaged phosphohydrolase activity of the G6Pase- α -V332L variant was 81% of wild-type activity, but that this difference was not statistically significant. We thank the reviewer for highlighting the missing statistical comment. In revision, we have

included a statement of the lack of statistical difference in Methods, under Generation of the huR83C mouse on page 24. Lines 14-16 and in the legend to Supplementary Fig. 2 of the revised manuscript.

Partial physiological rescue: We have shown that mammalian G6Pase- α proteins share 87-91% amino acid sequence identity and are physiologically equivalent (Chou et al. *Curr Mol Med* **2**:121-143, 2002; Chou et al. *Nat Rev Endocrinol* **6**:676-688, 2010). GSD-Ia is an autosomal recessive disease and requires two pathogenic alleles to exhibit a phenotype. Consistent with this the phenotype of the global *G6pc1*^{-/-} mice mimics the phenotype of human GSD-Ia. We have previously shown, by titrating the gene augmentation therapy, that reconstitution of ≥ 5 units, equivalent to $\geq 3\%$ of hepatic G6Pase- α activity, is sufficient for the treated *G6pc1*^{-/-} mice to maintain blood glucose homeostasis (Lee et al. *Hepatology* **56**, 1719-1729, 2012; Kim et al. *Hum Mol Genet*, **24**, 5115-5125, 2015; Kim et al. *Mol Genet Metab* **120**, 229-234, 2017, Kim et al. *Hum Mol Genet* **26**, 1890-1899, 2017). As such, we do not consider the 81% of wild-type hepatic G6Pase- α activity of the p.V322L variant impacts the current results. In this study, we showed that the BEAM-301-edited huR83C mice harboring higher phosphohydrolase activity manifest an identical clinical phenotype to the wild-type mice characterized by fasting hypoglycemia, hepatomegaly, nephromegaly, hyperlipidemia, hyperuricemia, lactic acidemia, and growth retardation (Fig. 4).

Mouse background: We agree with the reviewer that a study using mice in a pure background would be a preferred model and initially in our study of GSD-Ia, we generated and characterized the global *G6pc1*^{-/-} mice in three difference backgrounds: C57BL/6, 129S4/SvJaeJ, and mixed C57BL/6/129S4/SvJaeJ (C57BL/129). We showed that the *G6pc1*^{-/-} mice in the C57BL/6 background died within hours of birth and the *G6pc1*^{-/-} mice in the 129S4/SvJaeJ background survived only up to 24 hours after birth, indicating that the *G6pc1*^{-/-} mice in either the C57BL/6 or the 129S4/SvJaeJ background are not suitable models to study GSD-Ia. In contrast, the *G6pc1*^{-/-} mice in the mixed C57BL/129 background could survive for several days. These were the mice we used.

By implementing a glucose therapy, we were able to prolong the survival of the *G6pc1*^{-/-} mice in the mixed C57BL/129 background up to age 3 weeks, which enabled us to demonstrate they had a metabolic GSD-Ia phenotype.

The same was true for the generation of the *G6pc1*-R83C mice (Arnaoutova et al *Mol. Ther.* **29**,1602-1610, 2021). The *G6pc1*-R83C mice in the C57BL/6 background died within hours of birth. However, the *G6pc1*-R83C mice in the mixed C57BL/129 background survived better. Based on the poor survival noted above, we chose not to breed the *G6pc1*-R83C mice into the 129S4/SvJaeJ background, to minimize the use of laboratory animals. We have included this information in Methods under Generation of the huR83C mouse on page 24, paragraph 2 of the revised manuscript.

The reviewer asked whether all of the huR83C homozygous mice used in this manuscript were of mixed background, which could be a confounding factor as the mice may exhibit varying levels of the mixed background, resulting in differences in phenotype severity.

We agree that mice exhibiting varying levels of the mixed background could result in differences in phenotype severity. To minimize this, we used a strict breeding strategy making sure that all the huR83C homozygous mice used in this study were of mixed background consisting of approximately 50% C57BL/6 and 50%129S4/SvJaeJ. We have included the mating strategy in Methods under Generation of the huR83C mouse on page 24 lines 25-27 and page 25, lines 1-5 of the revised manuscript.

2. **It is surprising that no off-target analysis was conducted.** The authors did not even mention off-target effects in the entire manuscript. This omission is unacceptable. The authors should perform at least one unbiased whole-genome off-target analysis in a cell line or genomic DNA from the cell line.

Response: We have conducted off-target analysis using primary human hepatocytes from 3 donors and our results have identified only one intronic guide-dependent off-target site. Based on an integrated risk analysis, an edit at this off-target site is unlikely to impact the function of any associated genes. As suggested, we have included the off-target analysis study in Results under Off-target analysis on page 8, added the results in Supplementary Fig. 3, expanded our Discussion to address this analysis on page 20, paragraph 2, and included the methodology in Methods under Off-target editing analysis on pages 27 to 29.

3. **G6pc flox mice have been reported. Are there any studies investigating the knockout of the G6pc gene specifically in the liver or kidney? Corrective editing was only detected in the liver, not the kidney in this manuscript. Therefore, I am curious to know if there are any pathogenic physiological effects in mice with only kidney deletion of the G6pc gene.**

Response: Studies using liver-specific *G6pc1*-knockout (L-*G6pc1*^{-/-}) mice that survive to adulthood have been reported by Mutel et al. (*J Hepatol* **54**, 529-537, 2011) and our group (Cho et al. *PLOS Genet* **13**, e1006819, 2017).

Clar et al (*Kidney International* **86**, 747-756, 2014) have generated a kidney-specific *G6pc1*-deficient mouse line that retained 50% of wild-type renal G6Pase- α activity. These kidney-specific *G6pc1*-deficient mice did not exhibit a typical GSD-Ia phenotype. They survived to adulthood and presented only early renal perturbations after age 6 months.

We have also studied GSD-Ia nephropathy in a different model. We generated kidney-specific *G6pc1*^{-/-} mice lacking any renal G6Pase- α activity by treating 2-day-old global *G6pc1*^{-/-} mice with 10^{12} vp/kg of rAAV8-G6PC that targets the liver but not

the kidney (Lee et al, *BBA-Mol Basis Dis* **1870**,166874, 2023). The resulting K-*G6pc1*^{-/-} mice survive to adulthood without hypoglycemic seizures and manifest a renal phenotype mimicking that of the global *G6pc1*^{-/-} mice. We have shown that both the global *G6pc1*^{-/-} and K-*G6pc1*^{-/-} mice display renal fibrosis that is mediated by activation of the Wnt/ β -catenin/renin-angiotensin system axis (Lee et al, *BBA-Mol Basis Dis* **1870**,166874, 2023).

These findings are consistent with the differing functional roles of G6Pase- α in the liver and kidney. The role of liver G6Pase- α is to maintain blood glucose homeostasis. During meals, excess blood glucose is taken up and stored in the liver as glycogen. When fasting, such as between meals and overnight, the liver produces endogenous glucose via glycogenolysis and gluconeogenesis to release 75–80% of the required blood glucose (Gerich, *Diabet Med* **27**, 136-142, 2010; Alsahli and Gerich *Diabetes Res Clin Pract* **133**, 1-9, 2017). In contrast, the kidney stores little glycogen under physiological conditions. When fasting, the kidney primarily provides glucose via gluconeogenesis contributing 20-25% of blood glucose (Gerich, *Diabet Med* **27**, 136-142, 2010; Alsahli and Gerich *Diabetes Res Clin Pract* **133**,1-9, 2017). This is consistent with the ability of K-*G6pc1*^{-/-} mice to survive to adulthood without hypoglycemic seizures.

4. The authors observed a wide range of editing efficiencies. NB-high dosed mice exhibited a range in base-editing efficiency of 0.8% to 35%, while 3W-high dosed mice varied with a range in base-editing efficiency of 2.9% to 36.6%. The variability in the NB group could be attributed to technical challenges of tail vein injection, but retro-orbital injection for the 3-week group should be easier to perform. Do the authors have any insights into why there is so much variability?

Response: BEAM-301 was infused into NB huR83C pups via the temporal vein and into 3W huR83C mice via the retro-orbital sinus. The large variation in base-editing efficiency in NB infusion was not surprising because studies have shown that LNP-mediated delivery in neonatal mice may be impaired (Khoja et al. *Mol Ther Nucleic Acids* **28**, 859-874, 2022). In all our previous rAAV-G6PC1-mediated gene augmentation studies (reviewed in Chou & Mansfield *Frontiers Mol Med* **3**,1167091, 2023), we have routinely infused rAAV-G6PC1 to 2-week-old *G6pc1*^{-/-} mice via the retro-orbital sinus and have not observed big variations in efficacy. Presently, we can only speculate that the variability in base-editing efficiency could be caused by LNP-mediated delivery of the editing reagents.

5. What is the lowest editing threshold required to achieve therapeutic benefit for GSD-Ia patients?

Response: Since there is no study on the threshold of G6Pase- α activity required to achieve therapeutic benefit for GSD-Ia patients, we can only speculate the editing threshold based on our mouse studies.

In rAAV-G6PC1-mediated gene augmentation studies, we have shown that

reconstitution of ≥ 5 units, equivalent to $\geq 3\%$ of hepatic G6Pase- α activity, is sufficient for the treated *G6pc1*^{-/-} mice to maintain blood glucose homeostasis for 50-75 weeks (Lee et al. *Hepatology* **56**, 1719-1729, 2012; Kim et al. *Hum Mol Genet*, **24**, 5115-5125, 2015; Kim et al. *Mol Genet Metab* **120**, 229-234, 2017; Kim et al. *Hum Mol Genet* **26**, 1890-1899, 2017).

In this study, we showed that the efficacy of correcting the *G6PC1*-R83C variant correlates with the activity of hepatic G6Pase- α restored in the edited mice. Moreover, the edited mice harboring ≥ 3 units of hepatic G6Pase- α activity displayed an improved metabolic profile, sustained 24 hours of fasting, and survived through 1 year. To translate the mouse studies to human we need to account for the different body mass, relative size of the liver, and metabolic demands on blood glucose homeostasis in the human, so only clinical studies can address this. On a simplistic level we would hope that a similar 3% restoration of total liver G6Pase- α activity would be a good guide.

6. Did the authors check the AST/ALT levels in the treated mice?

Response: Aspartate aminotransferase (AST) and alanine aminotransferase (ALT) are serum biomarkers for liver diseases (Tamber et al. *Mol. Biol. Rep.* **50**, 7815-7823, 2023). Both AST and ALT have been reported to be elevated in human GSD-la patients (Szymańska et al. *Diagnostics (Basel)* **10**, 297, 2020). As suggested, we have analyzed serum AST and ALT levels in the BEAM-301-edited huR83C mice that live to 53 weeks of age. The normal ranges for mouse serum AST are reported to be 90.1 ± 8.1 to 293.4 ± 62.7 U/L, and for mouse serum ALT are reported to be 46.2 ± 5.6 to 239.5 ± 141.2 U/L (Otto, et al. *J. Am. Assoc. Lab. Anim. Sci.* **55**, 375–386, 2016).

Our results showed that serum levels of AST and ALT in all nine mice treated at 3 weeks of age with a low dose of BEAM-301 (3W-301L-53W) were in the normal range and statistically similar to that of their age-matched control mice. In the 19 mice treated at birth with the high dose (NB-301H-53W), 17 of the 19 mice had AST and ALT in the normal range, statistically similar to their wild-type (mR83) control mice. In the 2 other mice, which expressed 27 and 41 units of hepatic G6Pase- α activity, the levels of AST and ALT were high, AST at 360 and 370 U/L; ALT at 323 and 139 U/L. Neither mouse displayed any obvious metabolic abnormalities, although it has been reported that handling a mouse primarily by the body can elevate serum ALT (Swaim et al. *J Appl Toxicol* **5**, 160-162, 1985). It is possible that mouse handling during blood collection contributed to the elevated serum levels of ALT in these two mice. We have included the data in Results on page 17, lines 4-17 and Fig. 8c of the revised manuscript. We have also included the measurement of serum AST and ALT activity in Methods under Measurement of serum metabolites on page 32, paragraph 1 of the revised manuscript.

7. In the long-term study, no premature death was observed in the NB-301H-edited huR83C mice. The editing rates ranged from ~0.5% to 53%. The mouse with the lowest hepatic G6Pase- α activity was recorded at 2.4 units in the NB-dosed huR83C mice. As the authors pointed out, the lowest detection level of quantitation for the microsomal G6Pase- α assay is 2 units. Therefore, this particular mouse exhibited G6Pase activity at knockout levels. How did it survive for 53 weeks and overnight fasting? Similar findings were observed in the 3W-L-dosed huR83C mice, where one edited mouse exhibited only 2.6 units of hepatic G6Pase- α activity.

Response: We thank the reviewer for these comments, which highlight an inadequate explanation of our assay methods. In the phosphohydrolase assays reported, the background activity is non-specific phosphatase activity, estimated by pre-incubating permeabilized microsomes in 20 mM sodium acetate buffer, pH 5 for 10 min at 37 °C to inactivate the acid-labile G6Pase- α . This background, which varies from lysate to lysate, can range from 0.2 to 2 units of activity and is subtracted from the measured value before reporting our results. Of the values we report, the lower limit of quantitation is 2 units. We have revised the description of our assay in Methods under Phosphohydrolase assays on pages 30-31 of the revised manuscript.

The reviewer is correct that the two lowest hepatic G6Pase- α activity we measured, 2.4 units for one of NB-301H-53W and 2.6 units for one of 3W-301L-53W mice would appear to be at the knock-out levels. Measurements of liver microsomal G6Pase- α activity in 3-week-old untreated *G6pc1*^{-/-} mice varies from 0.5 to 2.2 units (unpublished results). The lowest level of activity that can support long term survival has not been defined precisely because it lies below or within the lower limit of quantification of the assay. However, the key proof of a functional level of G6Pase- α is the survival of the mouse. In prior experiments, no *G6pc1*^{-/-} mice has survived to 8 weeks of age, even under a glucose therapy.

In rAAV-G6PC1-mediated gene augmentation studies, we have shown that, in the absence of a glucose therapy, all twenty-five treated *G6pc1*^{-/-} mice expressing <5 units (0.8 to 4.1 units) of hepatic G6Pase- α activity survived up to age 60 weeks, maintained glucose homeostasis, and sustained 24 hours of fasting (Lee et al. *Mol Genet Metab Reposts* **3**, 28-32, 2015; Kim et al. *Mol Genet Metab* **120**, 229-234, 2017; Zhang et al. *Biochem Biophys Res Commun* **527**, 824-830, 2020). Ten of these mice expressed <2 units of hepatic G6Pase- α activity.

In the current study, glucose therapy was not provided to BEAM-301-dosed huR83C mice, as described in Methods under Animal studies on page 29, lines 22-24. Based on the above-observations, a 53-week survival of the NB-301H-edited huR83C mice expressing 2.4 units of hepatic G6Pase- α activity and the 3W-301L-dosed huR83C mouse expressing 2.6 units of hepatic G6Pase- α activity is not surprising. We have included this information in Discussion on page 21, paragraphs 3 of the revised manuscript.

8. The **summary provided in lines 399-401 is highly misleading**. The authors reported a 53-week survival rate of 69% among the 21 huR83C mice treated with 301L at age 3 weeks, claiming that all BEAM-301-dosed huR83C mice displayed a near-normal metabolic phenotype and were leaner and protected against age-related insulin resistance is inaccurate and misleading.

Response: We apologize for our oversight. In the revision we have stated on page 18, lines 5-8 “In summary, at age 53 weeks, all surviving BEAM-301-dosed huR83C mice displayed a near normal metabolic phenotype and were leaner and protected against age-related insulin resistance.”

Reviewer #2 (Remarks to the Author):

In this manuscript, Arnaoutova et al. present a straightforward, systemic, elegant, and experimentally sound study in which base-editing is used to correct glycogen storage disease type 1a in mice. To my knowledge, this work provides the first preclinical proof of concept for this state-of-the art methodology for the treatment of a monogenetic metabolic disease. Find below a number of points that, when addressed, in my view will further improve the quality of this manuscript.

Strong points in the manuscript:

- Major strengths of the study are that it uses a humanized model of GSD Ia (by insertion of the entire human G6PC1-R83C coding sequence and disabling native mouse G6pc expression), and that it uses a more efficient gene-editing tool (CRISPR-based base editing) as compared to the authors' previous studies (using either homologous recombination or a CRISPR/Cas9-based double-strand oligonucleotide insertion strategy)
- Clear description of the elaborate analysis of the different possibilities and eventual results from the administration of BEAM-301 in Figure 1.
- Clear and extensive description of results in both NB and 3-week old mice, wild-type or heterozygous, with low or high doses of BEAM-301, and analyzed at different time points (3 weeks (for NB only), 8 weeks, and 53 weeks). However, some questions and points for improvement remain (see below).

Response: We thank the reviewer for the critical reading and support of the paper and appreciate the comment that our study is “straightforward, systemic, elegant, and experimentally sound in which base-editing is used to correct glycogen storage disease type 1a in mice.” We have addressed the comments/suggestion raised by the reviewer as detailed below:

Major Comments

- The authors mention they monitor phenotypic correction through one year after base-editing-mediated correction of the mutated G6PC. However, liver tumor formation, one of the major long-term complications of hepatic GSD Ia in both humans and mice, usually only develop around the time and after end point of the study presented in this manuscript. Could the authors explain why did not follow-up for a longer period to assess prevention of liver tumor formation? In addition, why did the authors only include NB-301H and 3W-301L treated groups, and not also (the NB-301L and) 3W-301H groups?

Response: The two major aims of the current long-term (53-week) study were to demonstrate a functional efficacy and establish the long-term durability of BEAM-301-mediated base editing. The 53-week study was not designed to establish the threshold of hepatic G6Pase- α activity required for tumor prevention although we were interested in observing the tumor frequency. We have included this information in Results under Long-term correction of metabolic abnormalities of the huR83C mice on page 14, lines 22-26 and page 15, lines 1-2 of the revised manuscript.

Our experimental design was informed by the data derived in our prior long-term (50-70 weeks) rAAV-G6PC1-mediated gene augmentation studies. In these, we had shown that reconstitution of ≥ 5 units of hepatic G6Pase- α activity is sufficient for the treated *G6pc1*^{-/-} mice to maintain blood glucose homeostasis in the absence of HCA/HCC development (Lee et al. *Hepatology* **56**, 1719-1729, 2012; Lee et al. *Mol Genet Metab Reposts* **3**, 28-32, 2015; Kim et al. *Hum Mol Genet*, **24**, 5115-5125, 2015; Kim et al. *Mol Genet Metab* **120**, 229-234, 2017, Kim et al. *Hum Mol Genet* **26**, 1890-1899, 2017; Zhang et al. *Biochem Biophys Res Commun* **527**, 824-830, 2020). We had also shown that all twenty-five treated *G6pc1*^{-/-} mice expressing < 5 units (0.8 to 4.1 units) of hepatic G6Pase- α activity maintained glucose homeostasis and survived long-term. While prevention of HCA/HCC is important long term, the incidence of hepatic tumors in mice expressing < 5 units of G6Pase- α is low. For instance our rAAV-mediated G6PC1 gene augmentation study (Lee et al. *Mol Genet Metab Reposts* **3**, 28-32, 2015; Kim et al. *Mol Genet Metab* **120**, 229-234, 2017; Zhang et al. *Biochem Biophys Res Commun* **527**, 824-830, 2020), we had 25 mice titrated to express < 5 units of hepatic G6Pase- α activity and of those only four, expressing 1.5 – 2.2 units, developed tumors, while the other 6 with < 2 units of hepatic G6Pase- α activity did not develop tumors. So, a statistically significant statement about HCA/HCC risk requires significantly more mice than needed to demonstrate long term persistence of expression and glucose homeostasis. Such an HCA/HCC study also requires a fine titration of low G6Pase- α activity (< 5 units) that we have not yet explored using gene editing.

In our 8-week study, we established that while 50% of newborn (NB) mice died prematurely when dosed with the low dose (BEAM-301L), the high dose (BEAM-301H) provided 100% survival. We also established that the high dose reconstituted 5 to 173 units of hepatic G6Pase- α activity. This met the conditions we sought for the long-term NB-study.

In our 8-week study in huR83C mice dosed at 3-weeks of age, the low dose (3W-301L-8W) mice expressed from 6-64 units of hepatic G6Pase- α activity, while the high dose (3W-301H-8W) expressed 30 to 294 units. Since all the mice in the high dose exceeded the minimum G6Pase- α activity required for metabolic correction (5 units), we chose the low dose that was closer to the anticipated cutoff levels for HCA/HCC prevention in which any drop-in long-term expression could increase the opportunity to detect HCA/HCC.

For these reasons, the long-term studies with NB-301L and 3W-301H were not considered sufficiently informative to include in our study design. Our hypothesis also suggests that a very low number of mice will express < 5 units of hepatic G6Pase- α activity and develop hepatic tumors. Indeed, our 53-week study showed that one of the NB-301H-53W (n = 19) and one of the 3W-301L-53W (n = 9) mice expressed 2.4 and 2.6 units of hepatic G6Pase- α activity, respectively. Both mice appeared to be tumor-free.

Our results from the BEAM-301 editing experiments are consistent with our previous gene augmentation studies, showing long term functional durability of the restored G6Pase- α activity. Given this, we are now in a position to do long term studies using a statistically significant number of animals to make clearer statements about the ability of editing to prevent HCA/HCC. Since the primary concern in GSD-Ia is long term glucose homeostasis for survival, we believe these studies of long term, functional and physiological durability are major steps forward.

- Data hepatic (pre)-tumor incidence, or liver tumor markers at the 53-week endpoint are not included for all 9 mice of the 3W-301L-dosed huR83C group. As HCA/HCC is a common and severe complication observed in correct glycogen storage disease type 1a patients and mouse models, it is valuable to include such data to assess the effect of the treatment on HCA/HCC susceptibility.

Response: While we agree that HCA/HCC is a severe long-term complication for some GSD-Ia patients, the most pressing therapeutic need is to find a way to stabilize long term blood glucose homeostasis and prevent the damage of glycogen buildup in the liver for all patients. As indicated in our response to Comment 1, the current long-term (53-week) study was designed to demonstrate a functional efficacy and establish the long-term durability of BEAM-301-mediated base editing. This 53-week study was not designed to establish the threshold of normal hepatic G6Pase- α activity required for tumor prevention because we have not established the optimal conditions to obtain enough huR83C mice expressing <5 units of hepatic G6Pase- α activity for such a study.

In the current 53-week study, hepatic G6Pase- α activity in the nine 3W-301L-53W mice were 2.6, 11, 17, 35, 46, 59, 61, 70, and 179 units. Except for the mouse expressing 2.6 units of hepatic G6Pase- α activity, none of the other 8 are expected to develop hepatic tumors and histochemical analysis confirmed that none of the 3W-301L-53W mice developed HCA/HCC. It would not be statistically informative to measure pre-HCC markers such as AFP or CEA in the one mouse expressing 2.6 units of hepatic G6Pase- α activity.

- The authors state that their results support the development of BEAM-301 as a potential therapeutic for patients with GSD-Ia carrying the G6PC-R83C variant. Can the authors elaborate shortly if this only applies to patients homozygous for this variant, or also for patient where only one of the mutated alleles contains this variant

Response: GSD-Ia is an autosomal recessive disorder and individuals carrying only one mutant allele are phenotypically normal. If developed as a therapeutic, BEAM-301-mediated base editing would only be relevant for homozygous GSD-Ia patients carrying two copies of the p.R83C variant or compound heterozygous GSD-Ia patients carrying a p.R83C allele in combination with another pathogenic G6PC1 variant. We have revised the last sentence in Abstract to “The durable pharmacological efficacy of base editing in huR83C mice supports the development of BEAM-301 as a potential

therapeutic for homozygous and compound heterozygous GSD-Ia patients carrying the *G6PC1*-R83C variant.

- Line 92-93: “The prevalent pathogenic *G6PC*-R83C variant contains a single G>A transition mutation.” However, earlier in the abstract and in the remainder of the work the authors state the mutation is *G6PC*-g.247C>T, which suggest it is a C>T mutation. Are we correct to assume the authors in lines 92-93 refer to the opposing strand?

Response: Yes, lines 92-93 refer to the opposing strand because adenine base editors enable the programmable conversion of A•T to G•C in genomic DNA. In the revised manuscript, we have adjusted the text in Introduction on page 4, lines 22-25 to “The prevalent pathogenic *G6PC1*-c.247C>T/p.R83C variant contains a single G>A transition mutation on the complementary strand that can be targeted by the adenine base editors (ABEs) that enable the programmable conversion of A•T to G•C in genomic DNA, and in principle could be used to precisely correct this mutation²⁴⁻²⁹. In revision, we also included this information in the legend to Fig.1a.

- Lines 124-125: “the mR83 and mR83/huR83C littermates display indistinguishable wild-type phenotypes and, therefore, were used as controls”. Do the authors have data to prove that they indeed have indistinguishable phenotypes? We do agree, based on previous studies by the authors using murine GSD Ia models, that wild-type and heterozygous *G6pc*-deficient mice are similar in terms of key GSD Ia hallmarks such as survival rate, tolerance to fasting etc, but other data indicates that some hepatic parameters, such as (mild) glycogen accumulation, may still be present in mice with reduced but substantial remaining *G6PC* activities (PMID: 34157136).

Response: GSD-Ia is an autosomal recessive disorder and individuals carrying only one diseased *G6PC1* allele are phenotypically normal. However, GSD-Ia patients do display phenotypic variations and there is a lack of genotype-phenotype correlation (Chou et al. *Curr. Mol. Med.* **2**, 121-143, 2002; Chou et al. *Nat Rev Endocrinol* **6**: 676-688, 2010). Therefore, variations in hepatic *G6Pase*- α activity restored by base editing might contribute to the observed phenotypic heterogeneity in the treated mice as was observed by Rutten et al. (*Hepatology*. **74**, 2491-2507, 2021).

We have not looked in detail at subtle phenotypic differences between wild-type and heterozygous mice, but there are no gross differences in blood glucose homeostasis or long-term risk to HCA/HCC, the primary concerns for this study. Accordingly, the mR83 and mR83/huR83C littermates, which display indistinguishable wild-type phenotypes, were used as the controls.

- Do the authors have an explanation why BEAM-301 treatment does not result in increased *G6Pase* activity in the kidney cortex (Fig. 2, 4)? The authors speculate in lines 447-453 in the discussion, but do they have any suggestions for gene editing delivery tools more suitable for the kidney? What about long-term complications in the kidneys of treated mice after 53 weeks?

Response: As the reviewer noted, we did show that LNP-formulated BEAM-301 targets the liver but not the kidney, which is reminiscent of the results we obtained using the rAAV8-G6PC1 gene augmentation vector now in clinical trials. In the Discussion, we stated that restoration of hepatic G6Pase- α expression in the BEAM-301-edited mice improved nephromegaly and BEAM-301 might have the potential to alleviate renal dysfunction. However, in the absence of gene editing in the kidney, we also anticipate nephropathy will develop in aged BEAM-301-edited mice.

Rocca et al. have shown that retrograde renal vein injection of a rAAV9 vector efficiently targets the kidney cortex and medulla (*Gene Ther.* **21**, 618-628, 2014), suggesting that retrograde renal vein administration of BEAM-301 to the huR83C mice may be worth exploring. We have included this information in Discussion, on page 20, lines 24-27 and page 21, lines 1-8 of the revised manuscript.

- There appears to be a discrepancy in blood glucose results in NB-301H-3W mice presented in Figure 2C versus 3A, with glucose levels being significantly lower in these animals versus controls in Figure 3A only. The authors should discuss and explain this difference.

Response: In this study, we stated in Methods that mice were not subject to fasting before euthanasia and tissue collection. Since serum metabolites in the untreated and BEAM-301-edited animals were analyzed in the non-fasted state and had access to food ad libitum, their serum glucose levels could vary greatly depending upon the time of the study relative to their last food intake. Therefore, it is not unexpected that blood glucose levels presented in Fig.2c and Fig.3a differed. We have included this information in Methods under Animal studies on page 29, lines 22-25 and page 30, lines 1-2 of the revised manuscript. In revision, we also stated mice used in this study were in the non-fasted state throughout this manuscript.

- In Figure 4G, liver glucose levels are lower in NB-301H-8W, but not in 3W-301H-8W mice as compared to controls. Yet, the degree of G6P and glycogen accumulation in the liver is similar in these 2 groups. The authors should discuss these different phenotypes.

Response: Glucose entering the liver is phosphorylated by glucokinase to G6P (Rajas et al. *Metabolites* **9**, 282, 2019). In gluconeogenic organs, there are multiple competing pathways utilizing intracellular G6P, including: G6Pase- α -mediated glucose production; glycolysis; hexose monophosphate shunt; and glycogen synthesis. We have shown that hepatic G6Pase- α deficiency mediates a reprogramming of G6P metabolism which is reflected in the clinical manifestations seen in GSD-1a (Cho et al. *Biochem. Biophys. Res. Commun.* **498**, 925-931, 2018). However, perturbations of the various G6P utilizing pathways in the liver with variable levels of hepatic G6Pase- α activity have not been carefully investigated. The BEAM-301-edited mice displayed variable levels of hepatic G6Pase- α activity and endogenous glucose which would result in variable extent of G6P metabolic reprogramming, resulting in a lack of correction between levels of hepatic glucose levels and levels of hepatic

G6P/glycogen accumulation. It would be valuable if future studies analyzed the effects of the varied hepatic G6Pase- α activity on pathways of G6P utilization. As suggested, we have included this in Discussion on page 22, paragraph 2 of the revised manuscript.

- Data presented in Figure 8 show that after 53 weeks, 3W-301L-dosed huR83C mice are leaner and more insulin sensitive compared to control mice. Given that renal (and likely intestinal) G6PC activity remains uncorrected in these animals, it can be hypothesized that that (e.g. endocrine signals from) G6PC deficient kidney cells and/or enterocytes contribute to this lean, insulin-sensitive phenotype. The authors should consider and discuss this possibility.

Response: In rAAV-G6PC1-mediated gene augmentation studies, we have shown that rAAV-G6PC1-treated *G6pc1*^{-/-} mice restoring G6Pase- α activity only in the liver but not in the kidney and intestine are protected against age-related obesity and insulin resistance (Kim et al. *Hum Mol Genet*, **24**, 5115-5125, 2015). The authors showed that activation of pathways including hepatic carbohydrate response element binding protein signaling, NADH shuttle system, and the AMP-activated protein kinase/sirtuin 1/peroxisome proliferator-activated receptor- γ coactivator 1 α signaling, contribute to this lean, insulin-sensitive phenotype. It would be valuable if future studies analyze the roles of endocrine signals from G6Pase- α -deficient kidney cells and/or enterocytes that could also contribute to this phenotype. As suggested, we have expanded our Discussion on page 22, paragraph 1 of the revised manuscript.

Minor Comments

- The gene/protein name for human G6PC should be corrected to G6PC1

Response: As suggested, we have used *G6PC1* for gene/mRNA throughout the revised manuscript.

- In the abstract, the authors state “BEAM-301, lipid nanoparticles containing guide RNA and mRNA encoding a newly-engineered adenine base editor”. However, to my knowledge the term ‘guide RNA’ is often used in combination with gene editing using the CRISPR/Cas9 system, and I presume the authors actually mean a ‘repair template’ to correct the mutated G6PC. If this is correct, could the authors adjust this?

Response: The term “guide RNA” used in this study is correct as the RNA guides/directs the adenine base editors to the target. The phrase “repair template” is more appropriate to CRISPR/Cas9 homology directed repair.

- Lines 60-62: “In addition, dietary therapies do not address the underlying pathological processes and long-term complications, so hepatocellular adenoma/carcinoma (HCA/HCC) and renal disease still occur in metabolically compensated patients”. This is indeed true, but the authors fail to highlight that dietary compliance and associated

good metabolic control can at least reduce the incidence of these long-term complications (eg PMID 25308557, 28568353, 28612263), which may be good to highlight

Response: In Discussion, we have stated that “Studies have shown that good metabolic control improves renal function in human GSD-Ia patients (Okechuku et al. *J. Inherit. Metab. Dis.* **40**, 703-708, 2017; Dambaska et al. *Pediatr. Diabetes* **18**, 327-331, 2017). In revision, we have modified our statement in Discussion on page 20, line 27 and page 21, lines 1-3 to “Studies have shown that good metabolic control improves renal function and leads to regression of HCAs in human GSD-Ia patients (Beegle et al. *J. Inherit. Metab. Dis. Rep.* **18**, 23-32, 2015; Okechuku et al. *J. Inherit. Metab. Dis.* **40**, 703-708, 2017; Dambaska et al. *Pediatr. Diabetes* **18**, 327-331, 2017). Our results suggest that by improving metabolic control, BEAM-301 has the potential to alleviate renal dysfunction and regress preexisting HCAs.”

- Line 69: “many patients have pre-existing antibodies to the rAAV vector”; can the authors strengthen this claim by a reference?

Response: As suggested, we have included the relevant references (Calcedo et al. *J. Infect. Dis.* **199**, 381–390, 2009; Boutin et al. *Hum. Gene Ther.* **21**, 704-712, 2010) in the revised manuscript.

- Fig. 2E: label on y-axis does not seem correct (relative values is indeed true for the organ/body weight ratios, but not for the body weight data (which is in grams, an absolute unit))

Response: The reviewer is correct. We apologize for our oversight and have used Values (g or %) in the y-axis of Fig. 2e, Fig. 4d, and Fig. 5d of the revised manuscript.

- Fig. 3B: Hepatic glycogen accumulation is quantified as nmol/mg; however, the molar mass of glycogen depends on the extent of its branched structure. Can the authors explain what value was used for this / how the quantification was performed?

Response: Hepatic glycogen was determined using the Glycogen Assay Kit obtained from BioVision (K646-100 or ab65620) following the manufacturer’s instructions. Calculations of glycogen in nmol/mg were based on a glycogen molar mass of 666.5777 g/mol. This information has been included in Methods under Measurement of hepatic metabolites on page 32, lines 11-12 of the revised manuscript.

- Fig. 3C: H&E staining on liver of NB-301H-3W with correction to 33 units still shows a bit of vacuolopathy (enlarged hepatocytes) that usually indicates glycogen and/or fat accumulation, yet in Fig. 3C the authors showed that hepatic levels of these metabolites in 301H mice are completely restored to wild-type levels. Can the authors explain the yet remaining slight enlargement of the hepatocytes?

Response: Hepatic levels of glycogen and triglyceride in the 3-week-old control mice averaged 296.2 ± 27.8 and 43.2 ± 5.6 nmol/mg, respectively which were statistically similar to hepatic levels of glycogen and triglyceride the NB-301H-3W mice at 328.8 ± 36.1 and 42.3 ± 5.0 nmol/mg, respectively. Hepatic levels of glycogen and triglyceride in the mouse expressing 33 units of hepatic G6Pase- α activity were 376 and 33 nmol/mg, respectively. It is possible that the mildly elevated hepatic glycogen content in this mouse contributed to the minor vasculopathy associated with glycogen accumulation.

- The authors note in Fig. 4 a reduced editing variability and increased editing efficiency when performed at 3W rather than in NB mice. Do the authors have potential explanations for this? And how would this impact clinical translation? Eg correcting the genetic defect earlier in life would potentially benefit the patient more (especially right after birth when demand for gluconeogenesis is high) vs potential reduced editing efficiency

Response: Studies have shown that LNP-mediated delivery in neonatal mice may be impaired (Khoja et al. *Mol Ther Nucleic Acids* **28**, 859-874, 2022) which could account for the large variation in base-editing efficiency in NB infusion. While speculating on clinical outcomes, based on these initial mouse data is risky, it is certainly in the patients' interest to be treated as early in life as possible for a good outcome. Earlier treatment would lead to less initial glycogen accumulation in the liver, and less risk of long-term damage. Understanding the efficiency of transduction at different patient ages will be important as the therapy matures towards human patients. However, we feel these issues may be more appropriate to address in a review article than this initial proof of concept study.

- Please indicate more clearly in the legends the fed/fasting state in which the parameters were analyzed (especially relevant for interpreting serum metabolite levels in Figures 3-4); eg. It looks like data in Fig. 4F is from fed mice, but in order to interpret GSD Ia biochemical symptoms (eg fasting hypoglycemia, fasting hypertriglyceridemia etc) it is also important to analyze these parameters in a fasted state. When carefully reading the methods section this becomes evident, but as the nutritional state is critical in this disease context, we would like to advise to also state this more clearly in the figure legends.

Response: We have stated in Methods under Animal studies on page 29, lines 22-24 that the edited mice were not subject to fasting before euthanasia and tissue collection. In revision we also stated that "Since the untreated huR83C, BEAM-301-dosed huR83C, and control mice used in the study were in the fed state and had access to food ad libitum, their serum glucose levels could vary depending upon the time of study relative to their last food intake" on page 29, lines 24-25 and page 30, lines 1-2 of the revised manuscript.

As suggested, the information indicating that mice in the non-fasted state were used in this study was included in all relevant figure legends of the revised manuscript. The

exceptions are the fasting glucose test shown in Fig 2c, Fig 4c, Fig 5c, Fig 6c, and Fig 7c and the insulin tolerance test shown in Fig. 8b where the 3W-310L-53W and control mice were subjected to fasting for 4 hours before receiving an intraperitoneal injection of insulin.

We did not conduct studies in mice under fasting because the untreated huR83C mice could not tolerate fasting.

- Fig. 5: The authors note that of the 21 3-week-old 301L mice, 18 survived, yet G6Pase activity was only assessed in 8 of them; is there any specific reason?

Response: In Results on page 13, lines 21-23, we stated that 21 huR83C mice were treated with 301L at age 3 weeks. Eighteen of the 310L-dosed huR83C mice survived to age 8 weeks and 8 of the surviving 3W-301L-edited mice were used for the 8-week study (the 3W-301L-8W group). Among the remaining “10” 3W-301L-edited mice, one died at age 39 weeks and the other 9 mice were used for the 53-week study (the 3W-301L-53W group).

- Do the authors have a potential explanation for the marked lower hepatic TG levels in 3W-301L-53W mice compared to controls (Fig. 7G, 8D)?

Response: In this study, we showed that BMI values of the 3W-301L-53W mice were significantly lower than that of the sex-matched littermate controls (Fig. 8a), demonstrating that the 3W-301L-53W mice expressing G6Pase- α activity only in the liver but not in the kidney are protected against age-related obesity. Accordingly, hepatic triglyceride levels in the leaner 53-week-old 301L-dosed huR83C mice were markedly lower than their sex-matched littermate controls shown in Fig. 7g and Fig 8e.

- Fig. 8A: How is BMI calculated in mice? (especially the length parameter)

Response: The BMI value in mice was calculated by dividing the body weight measured in grams by the square of the body length measured in cm. The body length of each mouse was measured as the distance from the tip of the nose to the base of the tail. This information has been included in Methods under Animal studies on page 30, paragraph 3 of the revised manuscript.

Reviewer #3 (Remarks to the Author):

Response: Thank you.

Base-editing corrects metabolic abnormalities in a humanized mouse model for glycogen storage disease type-Ia (NCOMMS-24-19638A)

Reviewer #2 (Remarks to the Author):

The authors have satisfactorily addressed our comments and suggestions, with one minor issue remaining. The authors have, as requested, now adapted the human gene/protein name for G6PC1 from G6PC to G6PC1. However, to our knowledge the official gene name for mice is still G6pc (without 1). Could the authors adapt and make sure G6PC1 is used when referring to human, and G6pc when referring to mouse?

Response: As suggested, we have used *G6pc* throughout the revised manuscript.